# Septal wall synthesis is sufficient to change ameba-like cells into uniform oval-shaped cells in *Escherichia coli* L-forms
Masafumi Hayashi[1,5], Chigusa Takaoka[1], Koichi Higashi [2], Ken Kurokawa [2], William Margolin [3], Taku Oshima [4] ✉ & Daisuke Shiomi [1] ✉

A cell wall is required to control cell shape and size to maintain growth and division. However, some bacterial species maintain their morphology and size without a cell wall, calling into question the importance of the cell wall to maintain shape and size. It has been very difficult to examine the dispensability of cell wall synthesis in rod-shaped bacteria such as *Escherichia coli* for maintenance of their shape and size because they lyse without cell walls under normal culture conditions. Here, we show that wall-less *E. coli* L-form cells, which have a heterogeneous cell morphology, can be converted to a mostly uniform oval shape solely by FtsZ-dependent division, even in the absence of cylindrical cell wall synthesis. This FtsZ-dependent control of cell shape and size in the absence of a cell wall requires at least either the Min or nucleoid occlusion systems for positioning FtsZ at mid cell division sites.

It is very important for cells to control their size. Several mechanisms underlying bacterial cell size homeostasis have been proposed, including the critical size model, in which cell division initiates upon reaching a size threshold, and the constant extension model, in which cell division starts when cells reach a certain length prior to cell division[1,2]. Almost all bacteria, including *Escherichia coli*, are surrounded by a cell wall or peptidoglycan, which determines the cell shape and size and protects bacterial cells from turgor pressure[3]. However, it is unclear how cell wall synthesis is involved in cell size homeostasis. In addition, the models described above are based on studies of walled bacterial cells. However, several species, including *Mycoplasma* species, maintain a constant cell shape and size without a cell wall. Therefore, there may be a mechanism(s) that controls cell shape and size, even without a cell wall.

Bacterial cells generally exhibit two modes of cell wall synthesis, one for cell elongation, mediated by the Rod complex (the elongasome), and another for cell division, mediated by the divisome[3,4]. The Rod complex and divisome are composed of cytoskeletal proteins such as bacterial actin MreB and bacterial tubulin FtsZ, respectively, and penicillin-binding proteins (PBP2 and PBP3). The Rod complex interacts with the divisome, and this interaction is probably important for switching cell wall synthesis from elongation to division modes[5,6]. FtsZ polymers assemble at mid cell to form a ring-like structure, called the Z ring, along with many proteins including ZapA, PBP3 (FtsI), FtsW, and FtsN[3,7], which constitute the divisome. Divisome proteins are recruited to the Z ring in a specific order[7–11], with FtsZ

and ZapA being early divisome proteins, followed by later recruits such as PBP3, FtsW, and FtsN. Cell division is coupled with cell wall synthesis and is initiated by the activation of PBP3 and FtsW[12–14], which are septal peptidoglycan synthases. However, studies of the bacterial divisome have focused mainly on the divisomes of *E. coli* and *Bacillus subtilis*[7]. Therefore, it remains unknown whether the Z ring also works in wall-less bacteria, such as *Mycoplasma* species, which lack most divisome proteins other than FtsZ[15].

Cell elongation is associated with cell shape determination because cell shape is determined by the shape of the newly synthesized peptidoglycan cell wall[16]. Perturbation of several *E. coli* proteins, including MreB, a scaffold protein for the Rod complex, the transmembrane protein RodZ, and PBP2, results in abnormal cell shape[17–22]. Cell division is associated with cell size control since cells become filamentous when cell division is inhibited in *E. coli*. In addition, a regulatory model of cell size control has been proposed in which cells elongate to a certain length and then divide[2]. However, it has never been examined whether cell wall syntheses for elongation and division are necessary for cell size control as inhibition of cell wall synthesis inhibits cell proliferation.

The *ftsZ* gene is not only widely conserved in walled bacteria but also in most wall-less bacteria and most archaea whose cell walls are different from those in bacteria[23,24]. The *ftsZ* gene was acquired at a fairly early stage of evolution[25,26], but the actual role of FtsZ in bacteria without a cell wall is uncertain. JCVI-syn3.0, a synthetic bacterium created at the Venter Institute, lacks *ftsZ* and shows a heterogeneous cell morphology, suggesting poor

[1]Rikkyo University, Tokyo, Japan. [2]National Institute of Genetics, Shizuoka, Japan. [3]UTHealth-Houston McGovern Medical School, Houston, USA. [4]Toyama Prefectural University, Toyama, Japan. [5]Present address: Gakushuin University, Tokyo, Japan. ✉e-mail: taku@pu-toyama.ac.jp; dshiomi@rikkyo.ac.jp

control of uniform cell size maintenance[27]. JCVI-syn3 + 126, which is JCVI-syn3.0 with 7 genes added back, has a relatively uniform spherical shape compared to JCVI-syn3.0[27,28]. Interestingly, *ftsZ* is one of the 7 re-introduced genes. Therefore, in wall-less JCVI-syn3 + 126 cells, FtsZ may play a role in regulating cell shape and size without a cell wall. To accurately investigate the role(s) of FtsZ in regulating cell shape irrespective of the presence or absence of a cell wall, it is important to compare the function of FtsZ in cells of the same bacterial species with and without a cell wall. To this end, we compared *E. coli* cells with a normal cell wall with cells lacking a normal wall, called L-forms.

Bacterial L-forms, originally discovered in bacterially infected rats[29], have the ability to proliferate without a wall under certain conditions. Recently, L-form cells were found in patients with recurrent bacterial infections[30]. In humans with bacterial infections, the addition of antibiotics with cell wall synthesis inhibitory activity inhibits cell wall synthesis and causes some bacterial cells to switch from a walled state to a wall-deficient L-forms. This switch allows the bacteria to survive in the presence of antibiotics. Upon termination of antibiotic treatment, the infecting L-form cells resume cell wall synthesis and grow normally in a walled state, causing recurrent infections[30]. In the laboratory, the conversion to L-forms is induced by inhibiting cell wall synthesis through the addition of antibiotics such as penicillin and fosfomycin, or by deletion of genes involved in the cell wall synthesis pathway, under conditions of high osmotic strength and high concentrations of $Mg^{2+}$ [16,31,32]. In addition, anaerobic conditions promote the conversion to the L-form[33,34].

Whether FtsZ is required for the division of *E. coli* L-forms is controversial. D'Ari's lab showed that FtsZ is required for L-form growth in the presence of cefsulodin[35] whereas Errington's lab showed that cell division depends on membrane fluidity and is independent of the Z ring machinery (divisome) in the presence of fosfomycin[36,37]. If L-form division is indeed independent of FtsZ, then division would likely be driven by unregulated physical forces such as membrane blebbing and tubulation. This L-form-specific cell division switches to normal cell division, which is dependent on the Z ring machinery, by restarting cell wall synthesis[35,38]. It is still unclear whether the Z ring can control cell division, even in L-form cells, and how FtsZ functions in wall-less cells. In *B. subtilis* L-forms[32] or wall-less *Mycoplasma genitalium*[39], FtsZ forms filaments or clusters instead of discrete Z rings, suggesting that a cell wall is required for Z ring formation.

In walled *E. coli* cells, the position of the Z ring is determined by the Min system and nucleoid occlusion[40,41]. The Min system inhibits Z ring formation at the cell poles via pole-to-pole oscillation of the MinCDE complex[40,42]. Nucleoid occlusion inhibits Z ring formation over the nucleoid through the action of the DNA-binding protein SlmA[41]. Thus, the placement of the Z ring is negatively regulated by both the Min system and nucleoid occlusion. The shapes of L-form cells and their nucleoids are different from those of normal-walled cells; the L-form is ameboid and has multiple nucleoids[43]. No clear cell poles exist in ameboid L-form cells, and the nucleoid partitioning mechanism in L-forms may differ from that in walled cells.

As mentioned above, studying the relationship between the cell size-determining mechanism and cell wall synthesis for cell division is challenging because cell wall synthesis is essential for walled bacteria; therefore, it is difficult to observe how the loss of the cell wall affects cell size regulation by using walled bacteria. In addition, it has never been examined whether the Min and nucleoid occlusion systems could function in wall-less cells that lack uniform cell shapes. Therefore, in this study, we address these questions using *E. coli* L-forms, which are free from the constraints of cell wall synthesis, cell shape, and cell lysis. We demonstrate that the formation of the septal cell wall only, without the side wall, is sufficient to confer uniform cell shape. We also reveal that the Min and nucleoid occlusion-mediated regulation of Z ring formation and Z ring positioning are functional in wall-less L-form cells. Furthermore, at least one of the Min or nucleoid occlusion systems is required for the conversion from the FtsZ-independent division in wall-less cells to the FtsZ-dependent division in septal walled cells and hence to form a uniform cell shape.

## Results and Discussion
### Confirmation of the non-essentiality of FtsZ in *E. coli* L-form cells
FtsZ is not essential for the viability of L-forms of *B. subtilis*[32]. In *B. subtilis* L-form cells, FtsZ forms filaments instead of discrete rings[32]. However, the essentiality of FtsZ in *E. coli* L-form cells is controversial[35,36]. We first examined whether *E. coli* cells could form L-form colonies without FtsZ in our experimental condition, a high-osmotic medium plate (NA/MSM) containing fosfomycin (Fos), penicillin G (PenG), or cefsulodin (Cef). Fos inhibits the activity of MurA, which catalyzes the first step to synthesize peptidoglycan[44]. PenG inhibits the transpeptidase activity of penicillin-binding proteins (PBPs)[45,46]. Cef inhibits the transpeptidase activities of PBP1A and PBP1B[47]. Therefore, peptidoglycan is not synthesized in L-form cells in the presence of Fos; less peptidoglycan is synthesized in L-form cells in the presence of PenG than in L-form cells in the presence of Cef[48].

Since FtsZ is essential for the growth of walled *E. coli* cells, and therefore *ftsZ* gene cannot be deleted, we constructed a strain in which the expression of the *ftsZ* gene could be induced by sodium salicylate (NaSal) and checked whether L-form cells could grow under FtsZ depletion. RU2055 (FtsZ-depletion strain) cells, in which *ftsZ* expression can be controlled, formed colonies on NA/MSM plates with the inducer (Supplementary Fig. 1a, +NaSal, NA/MSM). On the other hand, without the inducer, this strain failed to form colonies as walled cells on NA/MSM plate without antibiotics (Supplementary Fig. 1a, -NaSal, NA/MSM), indicating that FtsZ was produced and depleted enough to support and inhibit growth in the presence and absence of the inducer, respectively. We confirmed protein levels of FtsZ by immunoblot using anti-FtsZ antibody and found that little was produced without the inducer (Supplementary Fig. 1b, ΔftsZ -). The amount of FtsZ produced from the plasmid in the presence of NaSal was higher than that of FtsZ produced from the chromosome. We also noticed that FtsZ produced from the plasmid showed double bands (Supplementary Fig. 1b). We previously used the same plasmid to produce FtsZ and did not detect such double bands of FtsZ in immunoblots with anti-FtsZ antibody[49]. However, in another paper, FtsZ was detected as double bands when glucose was added to nutrient-depleted conditions although the detailed reason for this was not discussed[50]. Therefore, FtsZ may be modified in some way under the specific medium or nutrient conditions. The strain could grow as L-form cells on NA/MSM plates containing antibiotics, regardless of the presence or absence of the inducer (Supplementary Fig. 1a, +NaSal or -NaSal, +Fos, +PenG, and +Cef), indicating that FtsZ is not essential for the growth of *E. coli* L-forms, as previously indicated[36].

We recently developed a system to monitor the conversion of walled cells to L-form cells and the growth of L-form cells in real-time by microscopy[48]. Using this system, we confirmed that *E. coli* cells could be transformed from walled to ameboid L-forms and could grow as L-forms, even under FtsZ-depleted conditions (Supplementary Fig. 2a-c and Supplementary Movies 1-6). We first counted the number of cells 4 hours after the addition of antibiotic (Fos, PenG, or Cef). Of those cells, we counted the number of cells that were able to divide up to 12 hours later. 21.2% (22/104 cells), 55.7% (39/70 cells), or 30.4% (38/125 cells) of WT L-forms divided while 22.0% (26/118 cells), 33.3% (36/108 cells), or 44.2% (53/120 cells) of ΔftsZ L-forms divided in the presence of Fos, PenG, or Cef, respectively. Although the percentage of cells that divided when converted to L-form by Fos is lower than with other antibiotics, these results also suggest that growth and division of L-forms do not require FtsZ as previously shown[36].

The phenotypes when the antibiotic was removed from the medium in which the cells were grown as L-form cells differed depending on antibiotics. Under FtsZ-depleted conditions, L-form cells grown with Fos lysed during the return to a walled state after removing Fos from the medium (Supplementary Fig. 2a ΔftsZ, and Supplementary Movie 2). L-form cells grown with PenG did not revert to rod-shaped cells but instead were abnormally elongated (Supplementary Fig. 2b ΔftsZ, and Supplementary Movie 4). This cell morphology is similar to a phenotype, called coli-flower, exhibited by a mutant strain isolated from *ftsZ* null L-form cells as a strain capable of growing as walled cells without *ftsZ*[36]. Unexpectedly, L-form cells grown with Cef showed the coli-flower phenotype at the beginning of re-

synthesizing peptidoglycan (Supplementary Fig. 2c Δ*ftsZ*, and Supplementary Movie 6). However, the "branches" of the cells became rod-like shaped but grew abnormally long (Supplementary Fig. 2c Δ*ftsZ*, red arrow), probably because they could not divide due to the lack of *ftsZ*.

WT L-form cells generated by the addition of Fos can revert to normal walled cells when Fos is removed, but all the Δ*ftsZ* L-form cells generated by Fos lysed during the reverting process (Supplementary Fig. 2a). These results suggest that FtsZ is required to revert to walled cells from cells completely lacking cell wall. The results resemble the observation that cell division precedes cell elongation during reversion from spheroplasts without peptidoglycan[51]. We also noticed that WT L-forms generated by Fos lyse in a significant percentage of cells upon resynthesizing the cell wall. In fact, when we measured how many of the L-forms present after 12 hours in antibiotics were able to revert to rod-shaped cells, only 3.0% (4/132) of the Fos L-forms were able to revert to rod-shaped cells. This is an extremely low percentage compared to the cells that reverted to rod-shaped cells from PenG L-forms and Cef L-forms (18.4%, 38/207 cells, and 21.5%, 37/172 cells, respectively). These results indicate that it is possible, but very difficult, to revert from a state of complete absence of a cell wall (Fos L-forms) to cells surrounded by a cell wall.

In our experimental system, the NB/MSM medium is constantly flowing (approximately 2.5 μL/h) and cell division could potentially occur as a result of shear forces due to flow. To demonstrate that division of L-forms occurs independent of medium flow, we filled the device with antibiotic-containing NB/MSM medium and then introduced the bacterial cells to induce conversion to L-forms while the medium flow rate was set to 0. However, it was not possible to completely eliminate the flow velocity in this system, which was about 0.1–0.3 μL/h. We observed that Z ring-independent divisions occurred under the conditions, suggesting that L-form cells can divide under almost no flow conditions (Supplementary Fig. 3a and Supplementary Movie 7). We counted L-form cells that divided without the medium flow rate and found that cell division occurred in 23.0% (50/217) of Fos L-form cells. This was higher than cells that divided with medium flow (9.9%, 17/141). Although we cannot exclude the possibility that medium flow may promote cell division of L-forms, medium flow is not necessarily required for L-form division.

## Z rings are assembled but fail to constrict in L-form cells

Since FtsZ was not essential in *E. coli* L-forms[36], similar to what was observed in *B. subtilis* L-forms[32], we predicted that, in *E. coli* L-forms, FtsZ might form non-ring filamentous structures similar to those observed in *B. subtilis* L-forms[32]. To visualize the Z ring, we fused sfGFP (super-folder GFP) with ZapA (denoted as ZapA-GFP), a component of the Z ring dependent on the Z ring for its localization. As expected, ZapA-GFP formed ring-like structures dependent on FtsZ in walled cells (Supplementary Fig. 3b). Surprisingly, ZapA-GFP formed ring-like structures in *E. coli* L-form cells, unlike in *B. subtilis* L-forms (Fig. 1a +Fos and Supplementary Movie 8), and ZapA-GFP rings were dependent on FtsZ in *E. coli* L-form cells (Fig. 1b, +Fos -FtsZ and Supplementary Movie 9). To confirm that the Z ring (ZapA) does indeed form ring structures rather than a filamentous structure, we analyzed the localization of ZapA-GFP in three dimensions. We confirmed that ZapA-GFP also localizes in a ring-like structure in L-form cells (Supplementary Fig. 3c and Supplementary Movie 10).

We then analyzed whether L-form cells divided where the Z ring is formed. In walled cells, the position of the Z ring coincided perfectly with the division position (359/359 divisions), as expected. In contrast, most of the L-form cells divided independently of the Z ring (Fig. 1a and Supplementary Fig. 3d), while the position of the Z ring coincided with the division position in only 9.5% of these L-form cells (16/169 divisions, Supplementary Fig. 3e), suggesting that L-forms can divide independently of the Z ring. We will discuss the result that 9.5% of the cells appear to divide at the Z ring later. To exclude the possibility that Z rings found in *E. coli* L-form cells were formed in the original walled cells and persisted in L-form cells, we examined whether ZapA-GFP rings could be generated de novo in L-form cells when FtsZ production was restored. In the absence

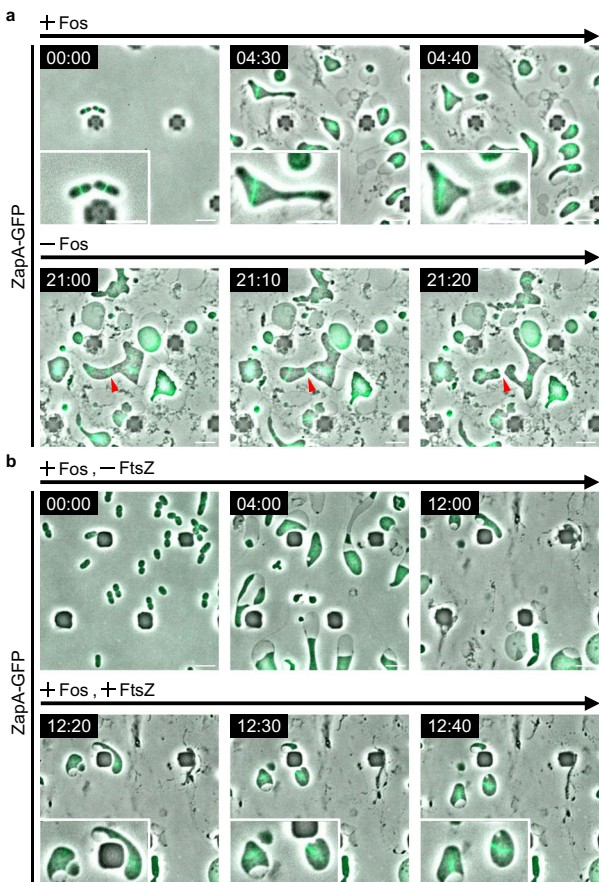

**Fig. 1 | Subcellular localization of ZapA-GFP in L-form cells. a** Time-lapse images of RU1125 (*zapA-sfGFP::cat)* cells grown in NB/MSM medium under anaerobic conditions containing Fos. Fos was removed from NB/MSM medium after 12 h. Images were taken every 10 min. Red arrowheads indicate cell division sites. Magnified images are also shown. Scale bars: 5 μm. **b** Time-lapse images of RU2057 (*zapA-sfGFP ΔftsZ::kan*/pWM2765) cells grown in NB/MSM medium under anaerobic conditions containing Fos. After 12 h, 10 μM NaSal was added to the NB/MSM medium. Images were taken every 10 min. Magnified images are also shown. Scale bars: 5 μm.

of FtsZ, ZapA-GFP diffused throughout the cytoplasm (Fig. 1b, +Fos, -FtsZ, and Supplementary Movie 9); however, after inducing *ftsZ* expression, ZapA-GFP re-assembled in L-form cells to form ring-like structures (Fig. 1b, +Fos, +FtsZ, and Supplementary Movie 9). These observations suggest that the Z ring can be generated in L-form cells and that Z ring assembly does not require a cell wall. Furthermore, when cell wall synthesis was resumed in L-form cells, the Z ring which had not constricted, began to constrict, and cell division occurred at the site where the Z ring was generated in L-form cells, suggesting that the Z ring was generated in a functional form in L-form cells (Fig. 1a, -Fos, Supplementary Fig. 4 and Supplementary Movie 8). This suggests that the Z ring is assembled normally in L-form cells but does not constrict, probably because all the components of the divisome are not assembled and/or cell wall synthesis was absent. In fact, a large number of divisome proteins must be assembled at the Z ring[3,9,52].

To investigate this question, we tested whether the late divisome proteins PBP3 (FtsI) and FtsN localize to the Z ring in L-form cells. PBP3, which is a monofunctional transpeptidase important for peptidoglycan synthesis at the division site, and FtsN, which activates the divisome and is the last protein to localize to it, are both essential for septal wall synthesis in walled cells[3,9,52]. To observe the localization of PBP3 and FtsN in L-form cells, we constructed plasmids carrying either msGFP2-PBP3 or msGFP2-FtsN. msGFP2 is a monomeric super-folder derivative of eGFP[53]. For simplicity,

msGFP2-PBP3 and msGFP2-FtsN are denoted as GFP-PBP3 and GFP-FtsN, respectively.

We found that in walled cells, GFP-PBP3 mainly co-localized with ZapA-mCherry, but a few dot-like foci of GFP-PBP3 were localized to the cell peripheries (Supplementary Fig. 5a). GFP-FtsN co-localized with ZapA-mCherry to the division site (Supplementary Fig. 5b). These results are consistent with those of previous studies[54,55]. In L-form cells, both GFP-PBP3 (Fig. 2a) and GFP-FtsN (Fig. 2b) formed ring-like structures and co-localized with ZapA-mCherry, as observed in walled cells, suggesting that a complete divisome is properly formed in L-form cells and is poised for constriction. FtsN contains a peptidoglycan-binding SPOR domain that facilitates its localization to division septa, but can also localize to Z rings independently of the SPOR domain through an interaction between its N-terminal cytoplasmic domain and the cell division protein FtsA, which itself is recruited to the Z ring[56,57] (Supplementary Fig. 6a). To confirm that the co-localization of FtsN and ZapA in the L-form we observed (Fig. 2b) was not dependent on cell wall, we examined the localization of FtsN lacking a SPOR domain (FtsN$^{\Delta SPOR}$) and a mutant composed only of the SPOR domain. Since the SPOR domain alone cannot be translocated to the periplasm, the TorA signal sequence, which is a transport signal to the periplasm, was fused to the SPOR domain (denoted as FtsN$^{SPOR}$) (Supplementary Fig. 6a, GFP-FtsN$^{SPOR}$). Consistent with previous studies[56], FtsN$^{\Delta SPOR}$ localizes primarily to the Z-ring in walled cells, whereas FtsN$^{SPOR}$ localized to the cell periphery (probably because of the binding to cell wall) but was enriched at the septum (Supplementary Fig. 6b). FtsN$^{\Delta SPOR}$ co-localized with ZapA-mCherry in the L-form as it did as in walled cells, whereas FtsN$^{SPOR}$ localized throughout the periplasm in L-forms, as expected (Supplementary Fig. 6c and 6d). The results suggest that FtsN localizes to the division septum in a Z ring-dependent manner even in the absence of cell wall.

## Reactivation of PBP3 changes the L-form cell division system from FtsZ-independent to FtsZ-dependent

Thus far, our data suggest that divisomes in *E. coli* L-form cells are fully assembled but fail to activate constriction because of the absence of a cell wall. Since it was shown that the PBP3-specific inhibitor aztreonam (Azt) inhibits Z ring constriction but does not inhibit Z ring assembly[58,59], we expected that if PBP3 regains its cell wall synthesis activity in L-form cells, the Z ring might become functional, even in L-form cells. As mentioned earlier, *E. coli* uses either the Rod complex or the divisome for its two modes of cell wall synthesis, which are mediated by PBP2 and PBP3, the specific transpeptidases for elongation and division, respectively[59,60]. PBP2 and PBP3 activities are specifically inhibited by mecillinam (Mec) and aztreonam (Azt), respectively[58,59]. Indeed, *E. coli* cells exhibited oval/round or filamentous shapes, respectively, in the presence of Mec or Azt in our microfluidics chamber (Supplementary Fig. 7a, 7b, and Supplementary Movie 11).

When walled *E. coli* cells were simultaneously treated with three antibiotics (Fos, Mec, and Azt), they were converted to ameba-like L-form cells that divided independently of Z rings (Supplementary Fig. 8a 4:00 and Supplementary Movie 12). Under this condition, cells were converted to L-forms by inhibition of Lipid II synthesis by Fos[37] and inhibition of PBP2 and PBP3 enzymatic activities by Mec and Azt, respectively. In these L-form cells, ZapA-GFP formed ring-like structures as observed in L-form cells converted by only Fos treatment (Fig. 1a and Supplementary Movie 8). Next, after conversion to L-form cells with Fos, Mec and Azt, only Mec or Azt was removed from the media, re-initiating cell wall synthesis for elongation or division in L-form cells. As expected, cells continued to grow as L-forms even though inhibition of PBP2 or PBP3 activity was released by the removal of Mec or Azt in the presence of Fos (Supplementary Fig. 8a, 8b, and Supplementary Movie 12). We also removed only Fos from medium containing Fos, Mec, and Azt. In the presence of both Mec and Azt, the cells proliferated as L-form cells (Supplementary Fig. 8c and Supplementary Movie 12). This suggests that inhibition of the activity of PBP2 and PBP3 by Mec and Azt, respectively, allows cells to grow as L-forms.

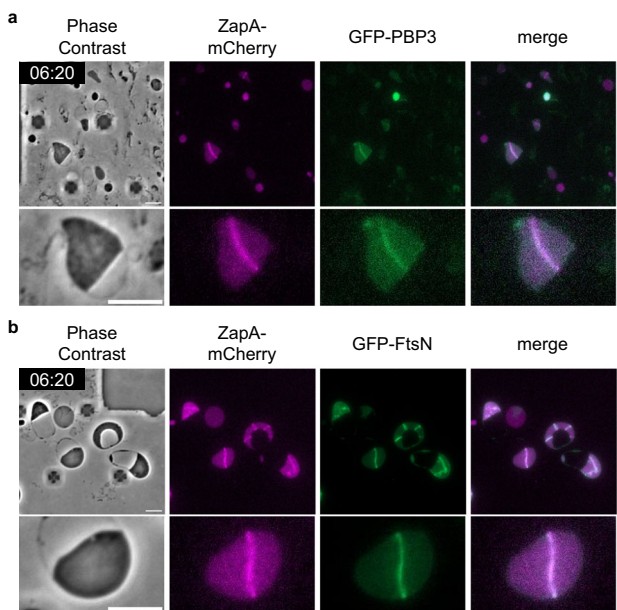

**Fig. 2 | Subcellular localization of late division proteins and ZapA-mCherry in L-form cells.** Images of RU2452 (*zapA-mCherry*) /pRU2282 (GFP-PBP3) (**a**) or pRU2281 (GFP-FtsN) (**b**) cells were grown in NB/MSM medium containing Fos and IPTG under anaerobic conditions. After conversion to the L-form, the growth conditions were changed from anaerobic to aerobic as mCherry could not be observed under anaerobic conditions. The magnified images are shown. Scale bars: 5 μm.

Next, after conversion to L-form cells with Fos, Mec, and Azt treatment, Fos and Mec were removed from the media, re-initiating cell wall synthesis catalyzed only by PBP2 in L-form cells. These cells, synthesizing the cell wall only for elongation (SWE cells), elongated and swelled in various directions, but the Z rings did not constrict (Fig. 3a +Azt and Supplementary Movie 13). This was probably because Azt inhibited the activity of PBP3, and Z ring constriction was specifically inhibited. In contrast, if Fos and Azt were removed from the medium, thus re-initiating cell wall synthesis catalyzed only by PBP3 in L-form cells, approximately 7 h after the removal Z rings started to constrict in cells synthesizing the cell wall only for division (SWD cells), triggering division (Fig. 3b +Mec and Supplementary Movie 14). This suggested that activation of PBP3 is sufficient to initiate Z ring constriction and cell division in SWD cells. Although it was previously thought that cell wall synthesis for division is triggered by the interaction between the Rod complex containing PBP2 and the divisome containing PBP3[5], PBP2-mediated synthesis of cylindrical cell wall is not required for constriction and division. This resembles the division of *E. coli* cells lacking *rodA* or *mreBCD* in which the activity of the Rod complex is inactivated or decreased.

To confirm whether cell wall synthesis is re-initiated only at the Z ring constriction in SWD cells, cell wall synthesis was visualized by GFP-FtsN$^{SPOR}$, which binds specifically to newly synthesized septal peptidoglycan[61]. After the removal of Fos and Azt, GFP-FtsN$^{SPOR}$ was localized in the periplasm in an ameba-like cell (Fig. 3c 19:00). However, at 19:30 and 20:00, cell constrictions were clearly observed in the cells and GFP-FtsN$^{SPOR}$ was localized at the constriction sites (Fig. 3c, Supplementary Fig. 9, Supplementary Movie 15), suggesting that the cell wall is synthesized at the constriction site. It should be noted that GFP-FtsN$^{SPOR}$ seemed to localize only at the constriction site in SWD cells while it localized at the constriction site and cell periphery in walled cells (Supplementary Fig. 6b), consistent with the idea that SWD cells synthesize cell wall only at division sites. These results also suggest that reactivation of PBP3 changes the division system of L-form cells from FtsZ-independent to FtsZ-dependent. Although the molecular mechanisms and signals responsible for Z ring constriction remain incomplete[62–64], recent studies have shown that septal

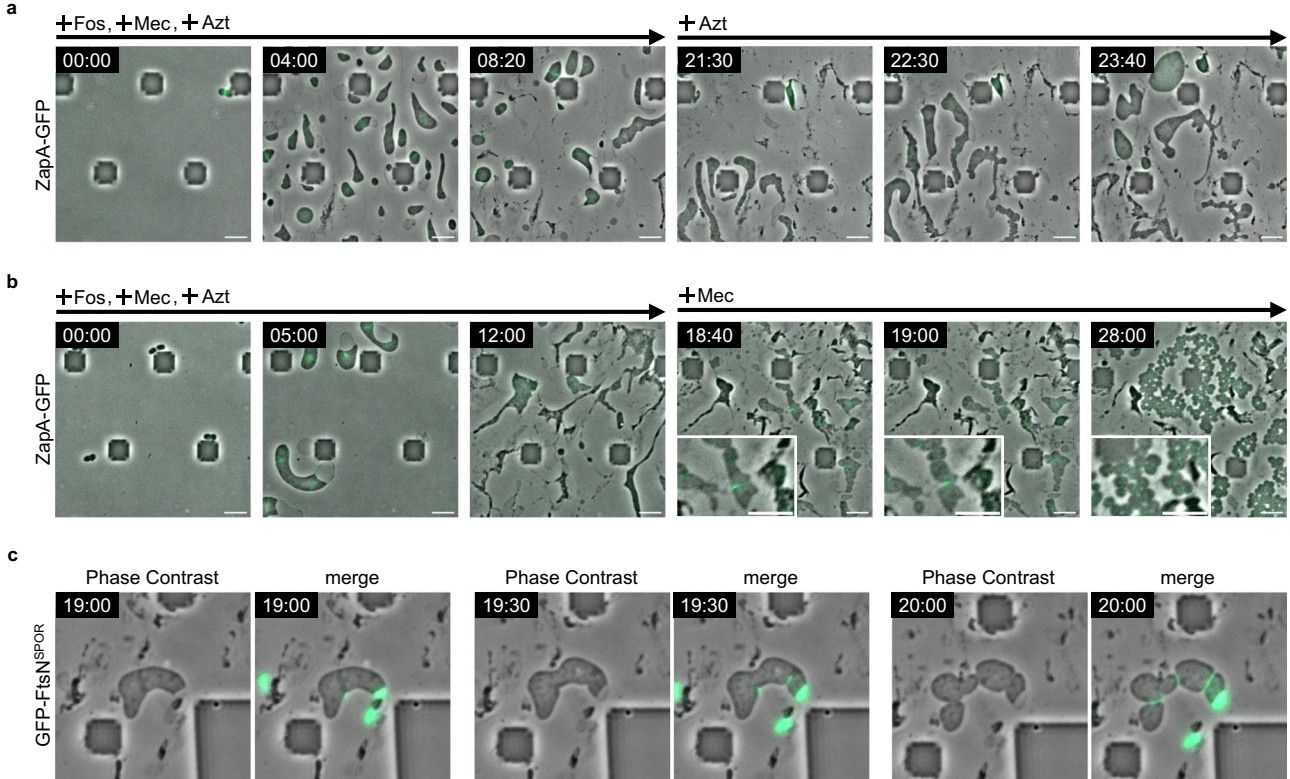

**Fig. 3 | PBP3 is important for Z ring constriction. a, b** Time-lapse images of RU1125 (*zapA-sfGFP::cat*) cells grown in NB/MSM medium containing Fos, Mec and Azt under anaerobic conditions. After 12 h, Fos and Mec (**a**) or Fos and Azt (**b**) were removed from the NB/MSM medium. Magnified images are also shown. Scale bars: 5 μm. **c** Time-lapse images of RU2452 (*zapA-mCherry*) cells carrying a plasmid encoding GFP-FtsN$^{SPOR}$ grown in NB/MSM medium containing Fos, Mec, and Azt. GFP-FtsN$^{SPOR}$ encoded on pRU2624 was expressed without IPTG. After 12 h, Fos and Azt were removed from the NB/MSM medium. After 16 h, the growth conditions were changed from anaerobic to aerobic. Scale bars: 5 μm.

cell wall synthesis is initiated by in vivo assembly and activation of FtsW and PBP3[14,65–67]. Using L-form cells in which the divisome is fully assembled but is blocked from constriction and septal synthesis, our results show that FtsZ-dependent cell division could be started only upon re-activation of PBP3. In walled-*E. coli* cells, the Rod complex containing PBP2 interacts with the divisome containing PBP3 to prepare for cell division[5]. However, our results show that FtsZ-dependent cell division occurs in SWD cells and does not require the synthesis of the cylindrical cell wall. This mode of division is similar to that of *E. coli* cells lacking *rodA* or *mreBCD* as described above.

What is the biological importance of Z ring formation in the *E. coli* L-form, in which the Z ring is not necessary for cell division? L-form cells have been linked to recurrent infections in patients treated with antibiotics[30,68–70]. The resistance to antibiotics through conversion to the L-form and re-conversion to normally growing walled cells from the L-form was observed in patients with recurrent infections[30]. Therefore, FtsZ-independent cell division in L-form cells may be a temporary adaptation to antibiotic treatment. If the Z ring could be formed in L-form cells surviving in a patient under antibiotic treatment, the bacterial cells would be able to resume cell wall synthesis and grow normally as walled cells through FtsZ-dependent cell division as soon as the antibiotics or cell wall synthesis inhibitors, such as lysozyme, are removed.

### L-form cells can divide to yield cells of homogenous sizes by reactivating PBP3

After the removal of Fos and Azt from the culture medium in which Fos, Mec, and Azt converted walled cells to L-form cells, the *E. coli* SWD cells converted gradually into uniform oval shapes during repeated divisions (Fig. 3b, 28:00, and Supplementary Movie 14). In those SWD cells, ZapA-GFP was localized at the mid-cell (Fig. 3b 28:00). This suggests that Z ring-dependent cell wall synthesis and cell division occur only at the mid-cell to

convert heterogeneously sized cells with ameboid morphology (L-form cells) into uniformly sized cells with an oval shape (SWD cells). This indicates that cell shape can be controlled solely by septal cell wall synthesis even if the cylindrical cell wall is not synthesized. This type of cell shape control is similar to, for example, those of *E. coli* Δ*rodA* cells and the coccoid *Staphylococcus aureus*. The cell division planes of *E. coli* Δ*rodA* and *S. aureus* are perpendicular to the previous division plane[71,72] while the division plane of WT-walled *E. coli* remains in a consistent perpendicular orientation. To investigate whether the orientation of the division plane of SWD cells is constant or changes with each division like Δ*rodA* and *S. aureus* cells, we analyzed SWD cell divisions in detail and found that the division plane of SWD cells is perpendicular to the previous division plane (Supplementary Fig. 10 and Supplementary Movie 14). This result indicates that the division mode of SWD cells resembles those of Δ*rodA* and *S. aureus*. Because nucleoid occlusion spatially regulates division sites in SWD cells like in *S. aureus*[73], nucleoid occlusion may regulate Z ring positioning (see below). Similarly, the wall-less synthetic bacterium JCVI-syn3.0 is irregularly shaped, but JCVI-syn3 + 126, in which several genes including *ftsZ* were re-introduced into this synthetic bacterium, became a uniformly sized coccus[27,28]. These results strongly indicate that cells can maintain a round or oval morphology even if they can divide in an FtsZ (or its functional equivalent protein) dependent manner without elongation like Δ*rodA* and Δ*mreB* cells.

### The Min system and nucleoid occlusion control spatial positioning of the Z ring in both L-form and walled cells

Z ring formation is negatively regulated by two systems, Min and nucleoid occlusion, in walled *E. coli* cells[40,41]. MinC protein, a component of the Min system, inhibits the assembly of Z rings at the cell poles. In Δ*minC* cells, the Z ring often forms at cell poles as well as at the cell center, resulting in

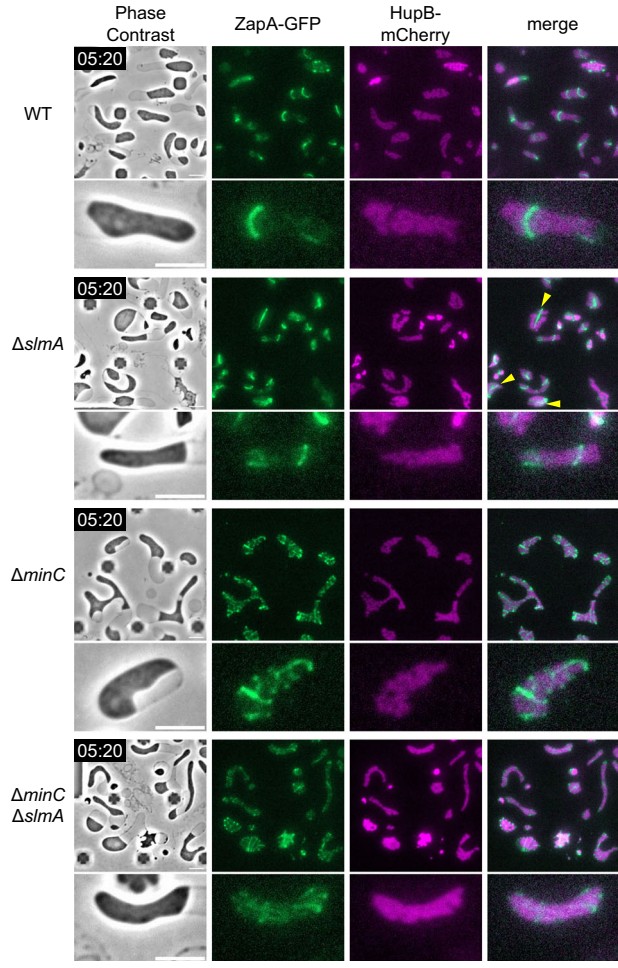

**Fig. 4 | Subcellular localization of ZapA-GFP in L-form cells lacking *minC*, *slmA*, or both.** Subcellular localization of ZapA-GFP and HupB-mCherry in the L-form cells. Cells (WT: RU1974, Δ*slmA*: RU2401, Δ*minC*: RU2400, Δ*minC* Δ*slmA*: RU2409) were grown in NB/MSM medium containing Fos under anaerobic conditions. After conversion to the L-form, the growth conditions changed from anaerobic to aerobic as mCherry could not be observed under anaerobic conditions. The magnified images are shown. The yellow arrows indicate the filament structure. Scale bars: 5 μm.

minicells, as previously reported[74] (Supplementary Fig. 11 Δ*minC*). Z ring formation over nucleoids is inhibited by SlmA bound to SlmA-recognition sequences in genomic DNA[75–77]. This negative spatial regulation is called nucleoid occlusion[75]. Z ring formation in Δ*slmA* mutant cells was indistinguishable from that in WT cells (Supplementary Fig. 11 Δ*slmA*), as previously reported[75]. Although the *minC* and *slmA* double mutant cells have been reported to exhibit synthetic lethality in LB media[75], we found that the strain was viable under high-osmotic culture conditions (NA/MSM), even as walled cells (Supplementary Fig. 11 Δ*minC* Δ *slmA*). Nonetheless, compared to the single gene deletion mutants, the Δ*minC* Δ*slmA* double mutant showed an exacerbated phenotype, with multiple Z rings and mislocalized rings (Supplementary Fig. 11 Δ*minC* Δ*slmA*).

We then examined the effect of the Min system and nucleoid occlusion on the Z ring formation in L-form cells. In wild-type L-form cells, ZapA-GFP was localized to the HupB-free (nucleoid-free) region (Fig. 4, WT), suggesting that nucleoid occlusion is functional in L-form cells. Unlike in Δ*slmA* walled cells, in Δ*slmA* L-form cells, some Z rings formed over the nucleoids (Fig. 4, Δ*slmA*), and some ZapA-GFP appeared to form filaments rather than ring-like structures (Fig. 4, Δ*slmA*; yellow arrowheads). Surprisingly, in Δ*minC* L-form cells, ZapA-GFP formed mesh-like structures (Fig. 4, Δ*minC*). However, ZapA-GFP did not localize over nucleoids,

indicating that nucleoid occlusion remained functional in Δ*minC* L-form cells. ZapA-GFP in Δ*minC* L-forms seemed punctate compared to WT or Δ*slmA* L-forms. This suggests that MinC plays a role in forming smooth filaments or that Z-ring components may form filaments with punctate if they do not form rings correctly. In addition, a mesh-like structure was also observed for GFP-FtsN in Δ*minC* L-form cells, suggesting that all components of the divisome were involved, even in the mesh-like structures (Supplementary Fig. 12 Δ*minC*). It should be noted that ZapA-GFP appears more punctate compared to the smoother mesh observed with GFP-FtsN in Δ*minC* L-form cells. This may be due to a slight overproduction of GFP-FtsN expressed from the plasmid, which appears to form smooth filaments.

It is well-known that MinC prevents the Z ring from assembling at cell poles and depolymerizes FtsZ filaments in vitro[78]. This inhibitory activity of MinC near the cell poles helps to restrict Z rings at midcell by preventing aberrantly localized FtsZ polymers, which are reflected by the mesh-like structures in L-forms in the absence of MinC. Consistent with this, it was previously shown that FtsZ was disorganized even in walled spherical *E. coli* cells lacking *minCDE*[79]. Recently, it was also shown that when FtsZ is encapsulated within lipid vesicles, it forms mesh-like structures, whereas when MinC is added, it forms a ring-like structure[80]. These results suggest that MinC plays not only a negative role but also a positive role in organizing the Z ring at midcell. It should be emphasized that in *E. coli*, MinC along with its ATPase partner MinD oscillates from one pole to the other, whereas in *B. subtilis* MinC and MinD are anchored to the cell poles and largely act to prevent Z rings from forming new division septa at the cell poles[11,81]. In addition, in *E. coli* SlmA interacts directly with FtsZ filaments to cause depolymerization[82], whereas in *B. subtilis* Noc, an important protein for nucleoid occlusion in *B. subtilis*, binds to DNA and the cell membrane, physically inhibiting FtsZ filament formation in the vicinity of the membrane[83]. This difference in MinC-MinD dynamics and the different functions between SlmA and Noc may account for differences in the ability of *E. coli* and *B. subtilis* to form Z rings in L-form cells. L-form cells of double mutants (Δ*slmA* Δ*minC*) showed an additive phenotype of Δ*slmA* and Δ*minC* single deletion L-form cells (Fig. 4, Δ*slmA* Δ*minC*). These results suggest that the Min system and nucleoid occlusion, used to determine the localization of the Z ring in *E. coli* walled cells, are functional in *E. coli* L-form cells. Although L-forms divide in a Z ring-independent manner, division at a Z ring was observed in 9.5% of cells. As nucleoid occlusion is involved in determining the position of Z-rings in L-forms, we speculate that division sometimes coincides with Z rings because these cells tend to divide in spaces between nucleoids anyway.

**Cell shape of SWD cells lacking *minC* or *slmA***

We showed that FtsZ formed mesh-like structures in Δ*minC* L-forms (Fig. 4). Next, we examined whether the Min system contributes to cell size regulation in SWD cells. Compared to wild-type (WT) SWD cells, the cell shape of Δ*minC* SWD cells appeared to be slightly heterogeneous, including anucleate mini cells and elongated cells. Measurement of the sizes of non-mini cells indicated that the Δ*minC* SWD cells were statistically significantly longer and wider than the WT SWD cells ($p < 0.05$, Fig. 5a, b), suggesting that cell size is regulated, at least in part, by the Min system in WT cells lacking cylindrical cell walls. However, the differences in the ratio between the length and width of WT and Δ*minC* SWD cells were not statistically significantly different (Fig. 5b), suggesting that the Min system controls the length and width of SWD cells synthesizing the cell wall only at mid-cell, but does not control overall cell shape and that SlmA-mediated nucleoid occlusion functions in Δ*minC* SWD cells.

The shapes of Δ*slmA* SWD cells and Δ*minC* SWD cells were also homogeneous like WT SWD cells. The ratio between the length and width of Δ*slmA* SWD cells and Δ*minC* SWD cells was comparable with that of WT SWD cells, although Δ*slmA* SWD cells and Δ*minC* SWD cells were longer and wider than WT SWD cells (Fig. 5b). Z rings were formed at mid-cell in the Δ*slmA* SWD cells, suggesting that the Min system is functional in the cells. Δ*minC* cells and Δ*slmA* cells showed similar shapes, indicating that nucleoid occlusion and Min system contribute equally to cell division in

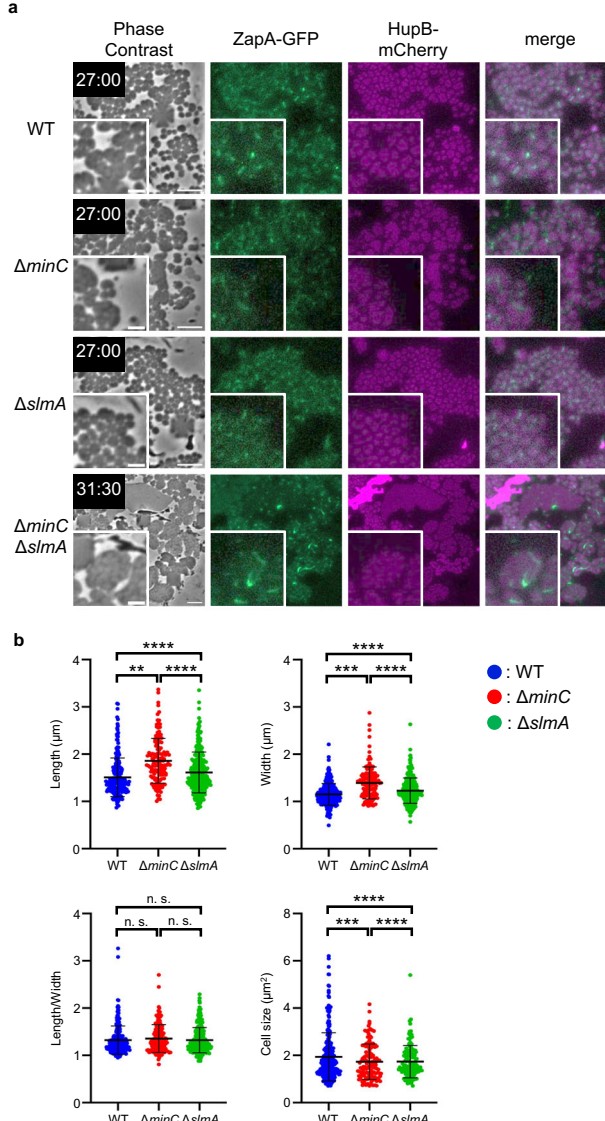

**Fig. 5 | Morphologies of SWD cells lacking Min and/or nucleoid occlusion systems. a** Images of RU1974 (WT), RU2400 (Δ*minC*), RU2401 (Δ*slmA*), and RU2409 (Δ*minC* Δ*slmA*) cells were grown in NB/MSM medium containing Fos, Mec and Azt under anaerobic conditions. After 12 h, Fos and Azt were removed from the NB/MSM medium. 12 hours after Fos and Azt removal, the growth conditions were changed from anaerobic to aerobic because mCherry could not be observed under anaerobic conditions. The magnified images are shown. Scale bars: 5 μm. **b** Morphological analysis of SWD cells. Cell length, width, and size were measured from pictures using MicrobeJ, a plug-in of ImageJ. Blue, red, and green dots indicate WT (*n* = 350 cells), Δ*minC* (*n* = 138 cells), and Δ*slmA* (*n* = 272 cells), respectively. Error bars indicate S.D. \*\**P* < 0.01, \*\*\**P* < 0.005, \*\*\*\**P* < 0.001 n.s.: not significantly different.

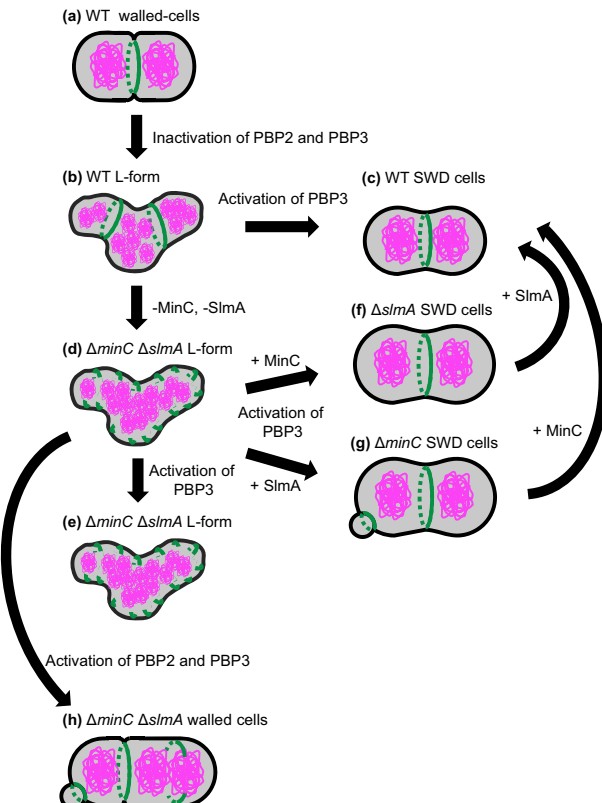

**Fig. 6 | Model for cell shape control and Z ring formation in SWD cells.** In WT walled-cells, FtsZ (green) formed the Z ring between the segregated nucleoids (magenta) (**a**). In the WT L-form cells, the Z rings containing all components of the divisome proteins were formed in the nucleoid-free region and could not constrict (**b**). WT SWD cells were divided into uniform oval cells using Z-ring by the activation of PBP3 (**c**). In the Δ*minC* Δ*slmA* L-form, the short green lines show incomplete FtsZ rings or filaments. Δ*minC* Δ*slmA* L-form cells failed to divide in an FtsZ-dependent manner after the activation of PBP3 (**d**, **e**). In Δ*slmA* SWD cells and Δ*minC* SWD cells, Z ring formation was determined by the Min system and nucleoid occlusion system, respectively. In Δ*slmA* cells, the Z ring was formed at mid-cell by using only the Min system (**f**). While in Δ*minC* cells, the Z ring was formed at cell-pole and mid-cell but not on the nucleoids. Constriction of the Z-ring at the cell poles produces minicells of SWD lacking a nucleus (**g**). In Δ*minC* Δ*slmA* cells, cells were initiated to divide and converted to rod-shaped cells by activation of PBP2 and PBP3 (**h**).

SWD cells. To our surprise, the Δ*minC* Δ*slmA* L-form cells started to deform into an oval shape and divide in an FtsZ-dependent manner after the removal of Fos and Azt, but they could not deform completely into uniform oval shapes. Instead, they seemed to return to ameba-like morphology as if they were L-forms (Fig. 5a, Supplementary Fig. 13 and Supplementary Movie 16). These results suggest that cells lacking a cylindrical cell wall cannot maintain a uniform cell size without both the Min system and nucleoid occlusion. Previously, it was shown that the Z ring localizes to the division site independently of the Min system, nucleoid occlusion system, and the Ter-linkage in walled *E. coli* cells[84]. Considering their data together with our data, it is possible that cell wall synthesis in the cylindrical portion may facilitate the localization of or stabilize the Z ring to the division site. In

other words, if there are systems that localize the Z ring to the right place in the cell, such as the Min system and/or nucleoid occlusion, FtsZ can maintain a uniform cell size without a cylindrical cell wall. These are summarized in Fig. 6. In WT walled-cells, FtsZ (green) forms the Z ring between the segregated nucleoids (magenta) (a). In the WT L-forms, the Z rings containing all components of the divisome proteins are formed in the nucleoid-free region but cannot constrict (b). WT SWD cells are divided into uniform oval cells using Z-ring by the activation of PBP3 (c). In the Δ*minC* Δ*slmA* L-forms, the short green line shows incomplete FtsZ rings/FtsZ filaments (d). Δ*minC* Δ*slmA* L-forms fail to divide in an FtsZ-dependent manner after the activation of PBP3 (e). In Δ*slmA* SWD cells and Δ*minC* SWD cells, the Z ring formation is determined by the Min system and nucleoid occlusion system, respectively. In Δ*slmA* cells, the Z ring is formed at mid-cell by the Min system (f). While in Δ*minC* cells, the Z ring is formed at cell-pole and mid-cell but not on the nucleoids. Constriction of the Z-ring at the cell poles produces mini-SWD cells without a nucleus (g). Δ*minC* Δ*slmA* walled-cells are initiated to divide and convert to rod-shaped cells by activation of both PBP2 and PBP3 (h). Δ*minC* Δ*slmA* L-form cells treated with Fos, Mec, and Azt, were able to revert to rod-shape when all antibiotics were removed (Supplementary Fig. 14 and Supplementary Movie 17). This confirms the importance of the cylindrical cell wall in

determining cell size and homogeneous shape as well as the cell wall at the division site, in the absence of nucleoid occlusion.

## Cell size control in wall-less bacteria

Cell size control is an important issue in bacterial physiology, and several models have been proposed[1,2,85]. However, these models are mainly based on experiments using walled cells, and it would be useful for understanding how primitive cells evolved to know more about regulation of cell size in wall-less bacteria, such as *Mycoplasma*. The synthetic bacterium JCVI-syn3.0 has the smallest artificial genome, which is based on the *Mycoplasma mycoides* genome[27,86]. JCVI-syn3.0, which lacks both a cell wall and *ftsZ*, has a heterogeneous morphology and grows slowly. JCVI-syn3 + 126 is a derivative of JCVI-syn3.0 in which several genes, including *ftsZ*, were put back into the genome. This strain has a relatively uniform cell size and round morphology[28]. It should be noted that it is unknown whether the division of JCVI-syn3 + 126 cells is FtsZ-dependent or not. Nevertheless, this result strongly suggests that FtsZ or FtsZ-dependent cell division can function as a regulatory system for cell size control, even in wall-less cells. This study also reveals that wall-less L-form cells require a system of nucleoid occlusion that determines the proper localization of the FtsZ ring for cell size control to change cell division from FtsZ-independent to FtsZ-dependent. Only when cells synthesize cylindrical cell walls is the nucleoid occlusion system not necessary to regulate the cell size uniformity.

We then speculated that FtsZ could function as a factor to maintain uniform cell size and increase growth efficiency in primitive cells lacking cylindrical cell walls. We re-examined the conservations of the *minC* and *slmA* genes and again clearly indicated that *minC* and *slmA* are absent in many bacteria, as is already known[87] (Supplementary Fig. 15), suggesting that *minC* and *slmA* were not acquired early during the evolutionary process, while *ftsZ* is highly conserved in bacteria and archaea. These findings reinforce the idea that the FtsZ-dependent cell division system is ancient and was in place before the Min and nucleoid occlusion systems evolved (Supplementary Fig. 15). If primitive cells acquired FtsZ before the acquisition of the cell wall, an alternative system to the modern nucleoid occlusion system might have served as the sole spatial regulatory system to control localization of the Z ring and to confer uniform cell morphology and size until these cells acquired the ability to elongate their cell wall. One candidate for a system to determine the localization of FtsZ is the nucleoid. FtsZ can localize between the segregated nucleoids even in walled *E. coli* (this study) and *B. subtilis*[88] cells lacking the Min system and nucleoid occlusion. In these cells, nucleoids may play a role in positioning of FtsZ. It is therefore plausible to assume that the nucleoid itself served the function of nucleoid occlusion in primitive cells.

## Methods
### Bacterial strains and growth medium

All strains were derivatives of *E. coli* K-12 and are listed in Table 1. BW25113 is a WT strain. Cells were grown in NB/MSM medium[32] comprising 2× nutrient broth (NB) (Oxoid, United Kingdom) mixed 1:1 with 2× magnesium-sucrose-maleic acid (MSM; 40 mM MgCl$_2$, 1 M sucrose, and 40 mM maleic acid, pH 7.0). When the cells were converted to the L-form on the plates, 0.75% agar was added (NA/MSM). Antibiotics were added at the following concentrations: 300 μg/mL PenG, 400 μg/mL Fos, 100 μg/mL Cef, 20 μg/mL Azt, and 10 μg/mL Mec. More detailed culture conditions and other information are clearly described in the appropriate sections.

### Strain constructions

Detailed methods for strain constructions are described below.

**RU2055 (Δ*ftsZ::kan*).** P1 lysate, prepared from WM2767 (Δ*ftsZ::kan*), was used to transduce Δ*ftsZ::kan* into BW25113 to yield RU2055.

**RU2057 (*zapA-sfGFP* Δ*ftsZ::kan*).** P1 lysate prepared from WM2767 (Δ*ftsZ::kan*) was used to transduce Δ*ftsZ::kan* into RU1429 to yield RU2057.

### Table 1 | Strains used in this study

| Strains | Relevant genotype | Reference |
|---|---|---|
| BW25113 | *rrnB* Δ*lacZ4787* HsdR514 Δ*(araBAD)567* Δ*(rhaBAD)568 rph-1* | 98 |
| SN1187 | MG1655 Δ*hsdR* Δ*endA* Δ*recA*(Δ2,820,759-2,821,785) | 91 |
| DH5α | F-, *deoR*, *endA*1, *gyrA*96, *hsdR*17(rk-mk +), *recA*1, *relA*1, *supE*44, *thi*-1, Δ(*lacZYA-argF*)U169, (Phi80*lacZ*ΔM15) | 99 |
| WM2767 | W3110 Δ*ftsZ::kan*/ pWM2765 (pKG110-*ftsZ*) | 49 |
| RU2055 | BW25113 Δ*ftsZ::kan*/ pWM2765 (pKG110-*ftsZ*) | This study |
| RU1125 | BW25113 *zapA-sfGFP::cat* | 6 |
| RU1429 | BW25113 *zapA-sfGFP* | 6 |
| RU1237 | BW25113 *hupB-mCherry::cat* | Lab stock |
| RU2057 | BW25113 *zapA-sfGFP* Δ*ftsZ::kan*/ pWM2765 (pKG110-*ftsZ*) | This study |
| RU1974 | BW25113 *zapA-sfGFP hupB-mCherry::cat* | This study |
| RU50 | Δ*minC::kan* | Lab stock |
| RU2332 | Δ*slmA::kan* | Lab stock |
| RU2400 | BW25113 *zapA-sfGFP hupB-mCherry::cat* Δ*minC::kan* | This study |
| RU2401 | BW25113 *zapA-sfGFP hupB-mCherry::cat* Δ*slmA::kan* | This study |
| RU2405 | BW25113 *zapA-sfGFP hupB-mCherry* Δ*minC* | This study |
| RU2409 | BW25113 *zapA-sfGFP hupB-mCherry* Δ*minC* Δ*slmA::kan* | This study |
| TH1165 | SB2/*pmCherry-kan* | Lab stock |
| RU2443 | BW25113 *zapA-mCherry::kan* | This study |
| RU2452 | BW25113 *zapA-mCherry* | This study |
| RU2462 | BW25113 Δ*minC* | This study |

**RU1974 (*zapA-sfGFP hupB-mCherry::cat*).** P1 lysate prepared from RU1237 (*hupB-mCherry::cat*) was used to transduce *hupB-mCherry::cat* into RU1429 to yield RU1974.

**RU2400 (*zapA-sfGFP hupB-mCherry::cat* Δ*minC::kan*).** A P1 lysate prepared form RU50 (Δ*minC::kan*) was used to transduce Δ*minC::kan* into RU1974 to yield RU2400.

**RU2401 (*zapA-sfGFP hupB-mCherry::cat* Δ*slmA::kan*).** A P1 lysate prepared form RU2332 (Δ*slmA::Kan*) was used to transduce Δ*slmA::kan* into RU1974 to yield RU2401.

**RU2409 (*zapA-sfGFP hupB-mCherry* Δ*minC* Δ*slmA::kan*).** A P1 lysate prepared from RU2332 (Δ*slmA::kan*) was used to transduce Δ*slmA::kan* into RU2405 to yield RU2409.

**RU2452 (*zapA-mCherry*).** cells producing ZapA-mCherry were constructed as follows: DNA polymerase KOD One (TOYOBO) was used for polymerase chain reaction (PCR). Genomic DNA from TH1165 was amplified using primers 2470/2453, and pKD4[89] was amplified using primer 1679/2452. The second PCR was performed using these PCR products as templates and the primers 1679 and 2470. The PCR product was introduced into strain BW25113 carrying pKD46[89] via electroporation. Cells were selected on L plates containing 25 μg/ml Kanamycin (Kan). The resulting strain (RU2443, *zapA-mCherry::kan*) was transformed with plasmid pCP20[89] using selection for Amp resistance (Amp$^R$) at 30 °C. The strain was then incubated at 42 °C in the absence of Amp, and colonies that grew were screened for the Amp-sensitive and Kan-sensitive phenotype at 37 °C. The resulting strain was designated as RU2452.

RU2405 (*zapA-sfGFP hupB-mCherry ΔminC)* and RU2462 (*ΔminC*). RU2400 and RU50 were transformed with plasmid pCP20 using selection for Amp^R at 30 °C. The strains were then incubated at 42 °C in the absence of Amp, and colonies that grew were screened for the Amp-sensitive, Kan-sensitive, and Cm-sensitive phenotype at 37 °C. The resulting strains were designated RU2405 (*zapA-sfGFP hupB-mCherry ΔminC*) and RU2462 (*ΔminC*).

## Plasmid constructions

Detailed methods for plasmid constructions are described below. The plasmids and primers used in this study are listed in Tables 2 and 3. To construct pTrc99A-Kan, *Pst*I fragment of pKD4 containing the Kanamycin resistance gene, which was blunted, was introduced into the *Sma*I fragments of pTrc99[90]. Plasmids encoding msGFP2-FtsN and msGFP2-PBP3 were

constructed using iVec[91]. To construct pTrc99A-Kan carrying msGFP2-FtsN and msGFP2-PBP3, three PCR products were amplified using pTrc99A-Kan as a template and primers 2289/2291 and 2290/2291, using synthetic DNA of msGFP2 as a template and primers 2292/2293 and 2292/2294, and using the purified BW25113 genome as a template and primers 2295/2296 and 2297/2298, respectively. The three PCR products were introduced into SN1187 cells using a modified TSS method[91]. Plasmids encoding msGFP2-FtsN^ΔSPOR and torA-msGFP2-FtsN^SPOR were constructed using In-Fusion. To construct pTrc99A-Kan carrying msGFP2-FtsN^ΔSPOR, PCR product was amplified using pRU2281 as a template and primer 2636/2637. To construct pTrc99A-Kan carrying *torA*-msGFP2-FtsN^SPOR, three PCR products were amplified using pRU2281 as a template and primers 2638/2639 and 2640/2641, using the purified BW25113 genome as a template and primers 2642/2643. One or three PCR products were combined with In-Fusion HD Cloning Kit (Takara Bio USA, United States) and introduced into DH5α cells using a CaCl_2 method.

## L-form conversion on NA/MSM plates

L-form conversion on NA/MSM plates was performed following a previously published method[48]. Cells were grown overnight in NB/MSM at 30 °C under aerobic conditions. The cell density was then adjusted to $OD_{600} = 1.0$ NB/MSM using Nanodrop One (Thermo Fisher Scientific, Waltham, MA, United States). Serial dilutions ranging from $10^{-1}$ to $10^{-5}$ by NB/MSM were performed, and 5 μL of the $10^{-2}$ and $10^{-5}$ dilutions were inoculated on NA/MSM plates with or without antibiotics, respectively. The plates were incubated at 30 °C for 3 days under anaerobic conditions using an Anaeropack (MGC, Tokyo, Japan). Then, the colonies, along with some agar, were picked, placed on a slide, and covered with a coverslip. Samples were observed under an inverted microscope (Axio Observer; Zeiss, Jena, Germany) equipped with an Objective Plan-Apochromat 100×/1.40 Ph3 (Zeiss) and filter sets 38HE and 63HE (Zeiss). Images were taken and processed using ZEN (Zeiss), Adobe Photoshop 2022 (Adobe, United States), and ImageJ (NIH, United States) software.

**Table 2 | Plasmids used in this study**

| Plasmids | Relevant genotype | Reference |
|---|---|---|
| pKG110 | pACYC184 derivative containing the *nahG* promoter, Cm^R | 100 |
| pWM2765 | *ftsZ* in pKG110, Cm^R | 101 |
| pTrc99A | P_lac/trc, Amp^R | 90 |
| pTrc99A-Kan | P_lac/trc, Kan^R | This study |
| pRU2281 | *msGFP2-ftsN* in pTrc99A-Kan, Kan^R | This study |
| pRU2282 | *msGFP2-ftsI* in pTrc99A-Kan, Kan^R | This study |
| pRU2623 | *msGFP2-ftsN^ΔSPOR* in pTrc99A-Kan, Kan^R | This study |
| pRU2624 | *torA-msGFP2-ftsN^SPOR* in pTrc99A-Kan, Kan^R | This study |
| pKD4 | *FRT-kan-FRT*, Kan^R, Amp^R | 89 |
| pKD46 | Lambda Red recombinase, Amp^R | 89 |
| pCP20 | yeast Flp recombinase gene, Cm^R | 89 |

**Table 3 | Primers used in this study**

| Name | Sequence |
|---|---|
| 2289 | GGCTCGCCGCCGGGGGTTGATCTAGAGTCGACCTGCAGGCATG |
| 2290 | GGACAGGTGGCAGATCGTAATCTAGAGTCGACCTGCAGGCATG |
| 2291 | CTCTCAGTGCTATCCATGGTCTGTTTCCTGTGTGAAATTG |
| 2292 | CAATTTCACACAGGAAACAGACCATGGATAGCACTGAGAGCCTG |
| 2293 | ACATAATCTCGTTGTGCCACGGATCCGCCACTGCCTCCAGTG |
| 2294 | GTTTTCGCCGCTGCTTTCATGGATCCGCCACTGCCTCCAGTG |
| 2295 | CTGGAGGCAGTGGCGGATCCGTGGCACAACGAGATTATGTACG |
| 2296 | GCCTGCAGGTCGACTCTAGATCAACCCCCGGCGGCGAGCCG |
| 2297 | CTGGAGGCAGTGGCGGATCCATGAAAGCAGCGGCGAAAACG |
| 2298 | GCCTGCAGGTCGACTCTAGATTACGATCTGCCACCTGTCCCCTCG |
| 1679 | CAGAGAAATTTTGTCTTCACGGTTACTCTACCACAGTAAACCGAAAAGTGCATATGAATATCCTCCTTA |
| 2452 | TGGACGAGCTGTACAAGTAGGTGTAGGCTGGAGCTGCTTC |
| 2453 | GAAGCAGCTCCAGCCTACACCTACTTGTACAGCTCGTCCA |
| 2470 | CGTTACTTGAACAAGGTCGCATCACCGAAAAAACTAACCAAAACTTTGAAGGTGGCTCTGGTGGCGGTTCTGGTGGCATGGTGAGCAAGGGCGAGG |
| 2636 | CTCTAGATCATTTTTTCTCCGCCGTCGGTTTTGG |
| 2637 | GGAGAAAAAATGATCTAGAGTCGACCTGCAGG |
| 2638 | TGGCGGATCCGAGAAAAAAGACGAACGCCGC |
| 2639 | CGTTATTGTTCATGGTCTGTTTCCTGTGTG |
| 2640 | AGCGGCGGAATTCATGGATAGCACTGAGAGCCTG |
| 2641 | CTTTTTTTCTCGGATCCGCCACTGCCTCCAGTG |
| 2642 | ACAGACCATGAACAATAACGATCTCTTTC |
| 2643 | TATCCATGAATTCCGCCGCTTGCGCCGCAGTCG |

### L-form/walled cells conversions in a microfluidic device

The L-form/walled cell conversion in the microfluidic device was performed following a previously published method[48]. Cells were grown to the early log phase ($OD_{600}$ = 0.35–0.45) in NB/MSM at 30 °C under aerobic conditions. A 50 μL aliquot of the cell culture was centrifuged, and the pellet was resuspended in 1 mL NB/MSM. Next, 50 μL of cell suspension was loaded into B04A microfluidic plates (CellASIC ONIX, Merck, United States). Microfluidic plate ceilings had a stairs-like structure with the following heights: 0.7, 0.9, 1.1, 1.3, 2.3 and 4.5 μm. In this study, only 0.7 μm was used for all of observations. The air in the microfluidic plates was replaced with $N_2$ using an $N_2$ gas-generating device (SIC, Tokyo, Japan). In L-form cells, mCherry fluorescence was observed under aerobic conditions because mCherry was not fluorescent under anaerobic conditions. Once the cells were converted to the L-form under anaerobic conditions, the L-form cells were viable under aerobic conditions for several hours. The microfluidic device was controlled using the software supplied by the manufacturer (Merck). The microfluidic plates were placed under a microscope and observed under an inverted microscope (Zeiss) equipped with Objective Plan-Apochromat 100×/1.40 Ph3 (Zeiss) and filter sets 38HE and 63HE (Zeiss). Images were taken and processed using ZEN (Zeiss), Adobe Photoshop 2022 (Adobe, United States), and ImageJ (NIH, United States) software. Microscopic images were analyzed using ImageJ (National Institutes of Health) and MicrobeJ software[92]. Graphs were created using GraphPad Prism10.

### Microscopic observation

Walled cells were grown to the log phase under aerobic conditions and mounted on 2% agarose on M9 medium. The cells were observed using an Axio Observer (Zeiss, Oberkochen, Germany), and the images were processed using ZEN (Zeiss) and ImageJ.

### SDS-PAGE and immunoblotting

Cells were plated on NA/MSM plates in the presence or absence of antibiotics. Colonies were suspended by NB/MSM media and suspension cultures were adjusted by the value of $OD_{600}$. SDS-PAGE and Immunoblotting were performed following a previously published method[93]. Anti-FtsZ antibody[93] as the primary antibody and horseradish peroxidase-conjugated anti-rabbit IgG antibody (Jackson ImmunoResearch) as the secondary antibody were used. The signal was detected on ImageQuant 800 (Cytiva) using ECL Select Western Blotting Detection Reagent (Cytiva).

### Phylogenetic distribution of proteins in bacterial genomes

Of the 254,090 bacterial genomes included in the Genome Taxonomy Database Release 202 (GTDB R202)[94], 4,326 genomes meeting two criteria were selected: 1) designated as GTDB representative strains (type strains or high-quality genomes), and 2) having complete genome sequences. Proteomes of these genomes were used as reference databases. As search queries, for the MinC protein, we used the HMM profiles registered in Pfam (PF03775 and PF05209)[95], and for the SlmA protein, we used *hmmbuild* to construct the HMM profile from the corresponding protein sequence in *E. coli* (NP_418098.4). To identify these three proteins in the 4,326 genomes, we used *hmmsearch* in the HMMER 3.2.1[96]. From the search results, a genome was considered to contain a protein when the hit e-value was 1e-8 or lower. To visualize the distribution of proteins in the bacterial phylogenetic tree, we used the phylogenetic tree file provided by GTDB R202 and performed tree annotation and rendering using Interactive Tree Of Life (iTOL)[97].

### Statistics and reproducibility

Statistical analysis was performed with GraphPad Prizm 10.2.2. The mean and SD of the data were used to create the graphs, and *p*-values were obtained by unpaired *t*-test. *p*-value < 0.05 was defined as a significant difference. All the experiments were repeatedly carried out and typical results are shown. Sample sizes are shown in Figure legend.

### Reporting summary

Further information on research design is available in the Nature Portfolio Reporting Summary linked to this article.

## Data availability

All relevant data supporting the findings are available within the paper and the Supplementary Materials. Source data used for generating the plots in Fig. 5b are available in the Supplementary Data file associated with the manuscript. Uncropped and unedited blot images are included in the Supplementary Material file. All other data and materials supporting this publication are available from the corresponding author upon reasonable request.

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

## Acknowledgements

We thank Dr. Yoshikazu Kawai (The University of Sydney) for his helpful suggestions and comments. We also thank all members of the Oshima and Shiomi laboratories for their helpful discussions and suggestions. This work was supported by the JST CREST (grant number JPMJCR19S5), Japan, to D.S., grant GM131705 from the National Institutes of Health, USA, to W.M., NIG-JOINT 57A2024 from National Institutes of Genetics, Japan, to T.O. and LA-2024-005 from Institute for Fermentation, Osaka (IFO), Japan, to T.O.

## Author contributions

M.H. conceived the project, performed experiments, analyzed, interpreted the data, and wrote the manuscript. C.T. performed experiments and analyzed and interpreted the data. K.H. and K.K. performed bioinformatic analyses, interpreted the data, and wrote the manuscript. W.M. interpreted the data and wrote the manuscript. T.O. and D.S. conceived the project, analyzed, interpreted the data, and wrote the manuscript.

## Competing interests

The authors declare no competing interests.
