## [Transparent Peer Review file · Communications Biology]

Septal wall synthesis is sufficient to change amoeba-like cells into uniform oval-shaped cells in *Escherichia coli* L-forms.

Corresponding Author: Professor Daisuke Shiomi

Version 0:

Reviewer comments:

Reviewer #1

(Remarks to the Author)

In this manuscript Shiomi and colleagues investigate the recovery of cells induced to become L-forms. They develop a process so they monitor the dynamics of cell shape as PG synthesis is inhibited and then allowed to resume (elongasome or divisome or both). Basically, they show that distorted L-form cells can recover more uniform shape and size by utilizing Z ring driven septal PG synthesis. Also, they show that Min and SlmA contribute to recovery by locating the Z ring near midcell. It is a nice demonstration that Z rings can form in amorphous blobs, and if PG synthesis can occur, result in cell division, eventually leading to more consistent size and shape.

Errington's lab has looked at recovery of L forms with and without FtsZ when PG synthesis is allowed to resume. With FtsZ cell shape is recovered (both elongasome and divisome are active) whereas without FtsZ (only elongasome active) cells can recover a rod shape but die due to the absence of division.

This is somewhat like growing cells deleted for the elongasome where only septal PG synthesis is occurring. In that case the Min system is known to be important (probably SlmA as well but I don't think it was known at the time and may not have been tested). Also, knocking out the Rod system generally is lethal although suppressors occur at a high frequency. It appears here that the media for growing L-forms probably suppresses such lethality so suppressors are not needed.

One cautionary comment though is that the authors seem to overstate their case due to probably unintended language. For example, on line 35 it is mentioned that control is exerted in the absence of a cell wall. This is not strictly the case though as septal PG synthesis is required. Thus, what is shown is that starting out with amorphous blobs recovery of a more uniform shape can occur utilizing the cell division machinery (i.e. septal PG synthesis driven by the Z ring). The title correctly summarizes the findings. I don't think the findings are unexpected, like I said it is like growing cells without the elongasome, but monitoring dynamics of the recovery reinforces what we know about cell division.

Also, Z rings do not appear (except for 9.5%) to be required for division in the absence of a cell wall as shown in ref 39. But what do the authors think is going on in the 9.5% of cells where FtsZ appears at constrictions. Is this mere coincidence?

During the recovery from inhibition of cell wall synthesis is it possible to estimate the efficiency of this process? It appears to be fairly efficient but hard to judge. What I am asking is how much cell mass is lost and unable to recover.

Line 57. Why not mention FtsW here since you bring in up a few lines later.

Line 64. "Work in other bacteria, including... This wording could be changed since wall-less bacteria lack most components

Line 83. I think the role of FtsZ is controversial since deleting FtsZ from the 19 genes added back did not affect cell size/shape much. Summarized in Trends in cell biology Volume 32, Issue 11, November 2022, Pages 900-907

Line 135. I don't think E. coli growing as an L-form without FtsZ is controversial. Ref 39 is very convincing that FtsZ is not required.

Sentence starting on line 141 is very long and could be split into two sentences.

Line 155-7. Ref 39 showed this was possible – i.e. for L-forms to grow without FtsZ

Line 176. Sentence starting with – These results... I cannot follow what you mean here.

The differences between the antibiotics is not clearly explained. Fos blocks at precursor stage, Pen only blocks transpeptidation so glycan chains can form (may or may not be stable), and with Cep, only the divisome is blocked.

Line 197. The 9.5% are interesting. Is FtsZ involved with these?

Line 302-3. Note point in Line 83 comment above.

Line 336. Corraling is the wrong word here. MinC prevents Z rings at poles but not at midcell.

Line 382. This is consistent with MreB cells grown with extra FtsZ or in minimal medium.

Ext data fig. 10. How sure are the authors about the conservation in this figure. For example, SImA is in the TetR family. Are the authors sure they are picking up true homologs and not paralogs? The names of the bugs are a bit faint in the figure.

Reviewer #2

(Remarks to the Author)

Examining the interplay between the cell size-determining mechanism and cell wall synthesis poses a challenge due to the fundamental role of cell wall synthesis in walled bacteria. In this study, Hayashi et al. utilized wall-less *E. coli* L-form cells and demonstrated that FtsZ-dependent septal cell wall synthesis alone is sufficient to transform heterogeneous cell morphology into predominantly uniform oval shapes. Additionally, they elucidated that at least one of the Min or nucleoid occlusion systems is requisite for the transition from FtsZ-independent division in wall-less cells to FtsZ-dependent division in septal-walled cells. The manuscript is lucidly articulated and logically structured. I would endorse its publication pending the resolution of the mentioned concerns.

Major Concerns:

1. The relationship between FtsZ ring constriction, septal cell wall synthesis, and cell division in L-form cells is not entirely clear. Previous work suggested that cell wall synthesis is the driving force for cell division in both *E. coli* (PMID: 26831086) and *S. aureus* (PMID: 36947615). Recent studies (PMID: 33495624, bioRxiv 515301) demonstrate that active septal cell wall synthesis can occur independently of fast FtsZ treadmilling dynamics. Therefore, it's crucial to determine whether FtsZ ring constriction triggers septal cell wall synthesis or if septal cell wall synthesis facilitates FtsZ ring constriction. Consequently, the statement regarding "FtsZ-dependent septal cell wall synthesis" should be carefully rephrased.
2. The authors state that FtsZ forms ring structures in *E. coli* L-form cells, which is distinct from *B. subtilis* L-forms. Given that a ring is a 3D structure not easily represented in 2D projections, could the authors employ 3D imaging techniques such as optical sectioning confocal microscopy, 3D-SIM, or 3D-SMLM to visualize the complete Z-ring structure?
3. The primary findings of this study are derived from observations of cell morphology. To bolster their conclusions, the authors should incorporate growth curves to provide evidence of cell division or lysis.

Minor Concerns:

1. It is advisable for the authors to conduct Western blotting to assess the residual amount of FtsZ under FtsZ-depletion conditions.
1. In Fig. S1, it would be beneficial to include growth curves of L-form cells after removing antibiotics instead of just showing some representative images.
2. The authors should compare the percentage of L-form cells with Z-rings coinciding with the division position under different antibiotic treatment conditions. If, as the authors suggest, "the constriction of the Z-ring requires cell wall synthesis," there should be a higher coincidence of Z-rings with the division position in cells treated with Cep compared to those treated with PenG or Fos.
3. Might the authors offer additional insights into the factors underlying the assembly of Z-rings in *E. coli* L-forms compared to their absence in *B. subtilis* L-forms, taking into account factors beyond the distinct MinC-MinD dynamics?
4. In Fig. 3a, it is noteworthy that a majority of cells lysed after 12 hours when treated with Mec and Cep. Could the authors measure the growth curve to demonstrate that the cells are still actively growing and dividing?
5. In Fig. 3b, the authors observe that *E. coli* SWD cells gradually convert into uniform oval shapes after the removal of Cep, and state "This indicates that cell shape can be controlled only by septal cell wall synthesis and not by the cylindrical cell wall." This conclusion may be too definitive, as the cylindrical cell wall likely plays a significant role in determining cell shape.
6. In Fig. 3c, could the authors also consider imaging ZapA-GFP to determine if the FtsZ-ring co-localizes with HADA signal?

7. Fig. 2 demonstrates that FtsZ (indicated by ZapA-mCherry) co-localizes with FtsN in L-form cells. However, in Fig. 4 and Fig. S7, both ZapA-GFP and GFP-FtsN display a mesh-like structure in Δ minC L-form cells. Notably, ZapA-GFP appears more punctate compared to the smoother mesh observed with GFP-FtsN. Does this imply that FtsZ is not co-localized with FtsN in Δ minC L-form cells?
8. Fig. 6 is a pivotal figure summarizing the model. It is mentioned only once incidentally in the main text. Could the authors add a sentence providing further description of this model in the main text?
9. Fig. S10 appears too small to discern details clearly. Consider enhancing the figure for better visibility.
10. The format of references should be consistent.

Reviewer #3

(Remarks to the Author)

Examining the interplay between the cell size-determining mechanism and cell wall synthesis poses a challenge due to the fundamental role of cell wall synthesis in walled bacteria. In this study, Hayashi et al. utilized wall-less *E. coli* L-form cells and demonstrated that FtsZ-dependent septal cell wall synthesis alone is sufficient to transform heterogeneous cell morphology into predominantly uniform oval shapes. Additionally, they elucidated that at least one of the Min or nucleoid occlusion systems is requisite for the transition from FtsZ-independent division in wall-less cells to FtsZ-dependent division in septal-walled cells. The manuscript is lucidly articulated and logically structured. I would endorse its publication pending the resolution of the mentioned concerns.

Major Concerns:

1. The relationship between FtsZ ring constriction, septal cell wall synthesis, and cell division in L-form cells is not entirely clear. Previous work suggested that cell wall synthesis is the driving force for cell division in both *E. coli* (PMID: 26831086) and *S. aureus* (PMID: 36947615). Recent studies (PMID: 33495624, bioRxiv 515301) demonstrate that active septal cell wall synthesis can occur independently of fast FtsZ treadmilling dynamics. Therefore, it's crucial to determine whether FtsZ ring constriction triggers septal cell wall synthesis or if septal cell wall synthesis facilitates FtsZ ring constriction. Consequently, the statement regarding "FtsZ-dependent septal cell wall synthesis" should be carefully rephrased.
2. The authors state that FtsZ forms ring structures in *E. coli* L-form cells, which is distinct from *B. subtilis* L-forms. Given that a ring is a 3D structure not easily represented in 2D projections, could the authors employ 3D imaging techniques such as optical sectioning confocal microscopy, 3D-SIM, or 3D-SMLM to visualize the complete Z-ring structure?
3. The primary findings of this study are derived from observations of cell morphology. To bolster their conclusions, the authors should incorporate growth curves to provide evidence of cell division or lysis.

Minor Concerns:

1. It is advisable for the authors to conduct Western blotting to assess the residual amount of FtsZ under FtsZ-depletion conditions.
1. In Fig. S1, it would be beneficial to include growth curves of L-form cells after removing antibiotics instead of just showing some representative images.
2. The authors should compare the percentage of L-form cells with Z-rings coinciding with the division position under different antibiotic treatment conditions. If, as the authors suggest, "the constriction of the Z-ring requires cell wall synthesis," there should be a higher coincidence of Z-rings with the division position in cells treated with Cef compared to those treated with PenG or Fos.
3. Might the authors offer additional insights into the factors underlying the assembly of Z-rings in *E. coli* L-forms compared to their absence in *B. subtilis* L-forms, taking into account factors beyond the distinct MinC-MinD dynamics?
4. In Fig. 3a, it is noteworthy that a majority of cells lysed after 12 hours when treated with Mec and Cep. Could the authors measure the growth curve to demonstrate that the cells are still actively growing and dividing?
5. In Fig. 3b, the authors observe that *E. coli* SWD cells gradually convert into uniform oval shapes after the removal of Cep, and state "This indicates that cell shape can be controlled only by septal cell wall synthesis and not by the cylindrical cell wall." This conclusion may be too definitive, as the cylindrical cell wall likely plays a significant role in determining cell shape.
6. In Fig. 3c, could the authors also consider imaging ZapA-GFP to determine if the FtsZ-ring co-localizes with HADA signal?
7. Fig. 2 demonstrates that FtsZ (indicated by ZapA-mCherry) co-localizes with FtsN in L-form cells. However, in Fig. 4 and Fig. S7, both ZapA-GFP and GFP-FtsN display a mesh-like structure in Δ minC L-form cells. Notably, ZapA-GFP appears more punctate compared to the smoother mesh observed with GFP-FtsN. Does this imply that FtsZ is not co-localized with FtsN in Δ minC L-form cells?
8. Fig. 6 is a pivotal figure summarizing the model. It is mentioned only once incidentally in the main text. Could the authors add a sentence providing further description of this model in the main text?
9. Fig. S10 appears too small to discern details clearly. Consider enhancing the figure for better visibility.
10. The format of references should be consistent.

Reviewer #4

(Remarks to the Author)

The authors studied cell shape and division phenotypes of *E. coli* after treatment with various drugs that interfere with all, or specific sets of, peptidoglycan (cell wall) synthases, in a nice experimental set-up that allows for long-time monitoring of growing walled or wall-less cells in microfluidic devices.

However, though treatment of cells with Fos, PenG, and maybe Cef are expected to lead to cells that are mostly wall-less or

L-form, there is no compelling reason to assume (as the authors do) that cells treated with both Mec and Cep will be without PG. Two major PG synthases (the preferred targets of Cef), PBP1A and PBP1B, are expected to still produce PG under those conditions. They may look amorphous and wall-less, but this is likely caused, at least in part, by the confined space in the microfluidic device. Thus, without convincing evidence showing otherwise, these cells should not be referred to as L-forms. As both the title, abstract and main conclusions of the manuscript are mainly based on the work with such (Mec + Cep treated) cells, from which then one of the drugs is often withdrawn (leading to the cells named SWE or SWD by the authors), the interpretation of the results by the authors is not convincing and likely wrong. Several other assertions by the authors are debatable as well.

Specific remarks

- 1) Lines 146-154. The authors find that wt and FtsZ-depleted cells can propagate as L-form on NA/MSM agar plates containing Fos, PenG, or Cef (Fig. S1a, as expected), and content that they do so in the microfluidic device as well (line 153).
 - a) Perhaps PenG or Cef-treated cells do so, but the pictures (Fig. S1b) and movies (1, 2, 7, 8) shown for Fos-treated cells are not convincing. Most cells seem to die/lyse. Remaining cell structures can change shape, but few, if any, actual 'division' events are evident before removal of the drugs. How do you know these cells actually propagate rather than a few cells barely surviving?
 - b) It is also interesting to note that most 'division' events in PenG or Cef-treated cells seem to correlate with cells that appear to be moving in a medium stream either hitting the support braces of the device, or being extended and 'pulled apart' within this stream (see movies, before drug withdrawal). This suggests that 'division' of these cells is driven by contact and/or shear forces within the device. In how far this is comparable to agar-grown L-form cells is unclear.
- 2) Line 171-172. The authors state that PenG-treated L-forms produce less PG than Cef-treated ones. As pointed out in ref 40 (but not here), however, the evidence for this is quite debatable as HADA incorporation itself may be less efficient in the former. This might also be the case for Fos-treated cells, by the way. The only way to know for sure is to actually extract and measure the amount of PG in the treated cells. This seems especially relevant to the Mec + Cep-treated cells (see below).
- 3) Lines 176-177. The statement 'These results suggest that FtsZ is required to trigger the first step of peptidoglycan synthesis.' is incompatible with the finding that delta-ftsZ cells can grow as walled coli-flowers (ref 39).
- 4) Lines 177-180. I don't understand this statement. In what way(s) are your results consistent with those findings in ref 41? Please explain to the reader.
- 5) Lines 182-192.
 - a) That E.coli L-forms can propagate without ftsZ was first shown by Mercier et al (Elife, 2014), and this reference should be added here.
 - b) Lines 183-185. The logic of this statement escapes me. Please explain to the reader.
 - c) Line 85 implies that B.subtilis L-forms do make Z-rings, while lines 189-190 imply the opposite. Which is it?
 - d) Also, please add the appropriate reference(s) concerning FtsZ localization in B.subtilis L-forms.
- 6) Lines 198-200. Perhaps this statement can be further solidified; did you see any constriction, without actual division, at any of the rings at all?
- 7) Lines 207-211. This statement is not supported by the data. The cell shown in Fig.1a is from 9 hr after Fos withdrawal. How likely is it that this ring persisted for 9 hr?
- 8) Lines 223-224.
 - a) A few 'what' were localized to the cell pole?
 - b) Also, I don't see any polar GFP-PBP3 in fig. S3a. There is a focus at a quarter position.
- 9) Lines 226-229. Fig. 2b.

The localization of GFP-FtsN looks pretty sharp and it would be interesting to know if this depended on its N-terminal and/or C-terminal domain, which are weak and strong localization determinants, respectively, in walled cells. As the C-terminal SPOR domain binds septal PG, it is expected to not be required for this localization, but only if these cells truly made no PG (see also point 2 above).
- 10) Lines 241-244. Cells treated with Mec and Cep
 - a) How do you know that these cells divided independently of the Z-ring? Did you do the same experiment with FtsZ-depleted cells? Please explain the evidence for this statement to the reader.
 - b) Are these true L-form cells? PBP1A and 1B are still expected to be active and produce PG. This is actually an important point. Wouldn't these cells just form large walled spheres in liquid, if they were not trapped by the low ceiling in the microfluidic device? And, if not, why not?
 - c) A related question is : What are the phenotypes of cells if they are treated with only Mec (or Cep) in this device and medium?
 - d) Why did you choose such high concentrations of Mec and Cep (300 microgram/ml, line 420)? At this concentration neither antibiotic is likely very specific for their intended targets anymore. Did you try lower concentrations?
 - e) Also, why did you choose cephalixin? Aztreonam is more specific for PBP3, and 2-3 microgram/ml is typically sufficient

to inactivate it.

11) Lines 249-251. The SWE cells are said to elongate (and swell). I actually see very little elongation in the sense of rod-shape elongation in Fig. 3a or movie 10. Why don't these cells revert to typical filament formation? Does the high Cep concentration perhaps also affect PBP2 activity?

12 Lines 253-258. SWD cells.

a) The conclusion that PBP2 activity is not needed for cell constriction is certainly not new. It has long been known that none of the rod-system proteins, including PBP2, are strictly required for cell constriction.

b) However, at high growth rates and/or rich medium, rod-system mutants typically form (very) large non-dividing walled spheres unless the cellular level of FtsZ is modestly elevated, in which case they can propagate as smaller dividing walled spheres. One surprising aspect of Fig. 3b and Movie 11, is that the SWD spheres (or 'ovals') at the end of the experiment are so very tiny, especially compared to the size of the rod-shaped cells at the beginning of the experiment. Why are they so tiny?

c) What is the estimated doubling time of wt cells without drugs in this device and medium?

d) If you grow an *mrda* (no PBP2) mutant under these anaerobic conditions, are they also this tiny?

e) Do these growth conditions perhaps elevate the cellular FtsZ concentration?

13) Line 259. The HADA experiment does not confirm that 'cell wall synthesis was re-initiated only at the Z ring constriction in SWD cells'. Brighter HADA staining at division sites is also seen in wt cells and does not mean PG synthesis is exclusive to these sites. In fact, one expects PBP1A and 1B to produce PG all along the periphery of the SWD cells. See point 10 as well.

14) Lines 271-272. The authors conclude: ' However, our results show that FtsZ-dependent cell division occurs in L-form cells and does not require the synthesis of the cylindrical cell wall.'

I disagree, because I don't believe these are L-form cells, but rather comparable/equivalent to walled rod-system mutant cells (such as *delta-mrda*). See points 10 and 12, above.

15) Lines 284-306, Fig. S5, Movies 11 and 12.

a) This whole section reinforces my argument (points 10, 12-14, above). These SWD cells behave like a *rodA* (*mrdB*) mutant because they basically are equivalent, except that these are essentially wt cells with PBP2 inactivated (so, more like a *mrda* mutant, but the phenotype is similar).

b) Just take wt cells, put them in the device and add medium with Mec. You'll end up with the same 'oval' cells. See point 10c as well.

16) Fig. S6. I assume growth conditions here were aerobic? Please state in the legend.

17) Lines 308-351. The results in this section are nice but, as indicated by the authors, not very surprising given what we know about FtsZ and the Min and SlmA systems, and previous results with *mre* mutants and lipid vesicles.

18) Lines 363-364. I don't believe SWD cells produce PG only at midcell as stated here and elsewhere. See points made above.

19) Lines 374-384. *delta-minC delta-slmA* SWD cells, Fig.5a, Fig. S8, Movie 13.

a) I predict that you'll see the same phenotype if you just grow the RU2409 cells in NB/MSM medium in batch culture (as in Fig. S6) or in the microfluidic device, but now with the addition of Mec. The many filamentous FtsZ structures that are allowed to form at random positions in the resulting walled spheres (batch) or squashed walled spheres (microfluidic device) in the absence of both MinC and SlmA compete with each other for division proteins such that assembly of a complete/functional ring is suppressed. Quite similar to what happens when *delta-minC delta-slmA* cells are grown in rich medium, and cells filament.

b) Oddly, Fig.S9 and Movie 14 are not mentioned/discussed. What is the point of this data?

20) Fig.6 is also ignored in the text.

a) What is the point of the figure?

b) One of the (f) panels is mislabeled.

21) Lines 183-500. Please indicate what ceiling height in the microfluidic device was used (0.7 micron throughout?)

22) Genetic descriptions of strains and/or strain names in the legends of figures and movies are often imprecise. Please check and correct.

Version 1:

Reviewer comments:

Reviewer #1

(Remarks to the Author)

This revision is improved. The main conclusion is supported by additional experiments. The conclusion is that Z rings can

form in amoeba-shaped E. coli L-forms and a complete divisome is assembled (FtsN is localized but SPOR is not-indicates a complete divisome is formed but not active). However, the Z ring does not constrict in the absence of PG synthesis. If septal PG synthesis occurs (PBP3 active), then division occurs in an FtsZ-dependent manner and amoeba-like cells are eventually converted to a more uniform coccal-shape. It is already known that rodA or mreBCD cells grow as coccal-shaped cells under slow growth conditions and that Min has an important role. The additional information here is that Z rings (and complete divisomes) form in the amoeba-like L-forms but don't constrict. However, if septal PG synthesis is allowed to occur, they constrict.

These experiments don't answer some of the big questions that are asked by the authors such as the role of FtsZ in wall-less cells. It is still not clear what the role of FtsZ is in wall-less mycoplasma. Clearly FtsZ has evolved in E. coli to be coupled to septal PG synthesis.

Some suggestions for improvement:

Line 47. Pressures does not need to be plural.

Line 51. "control" should be "controls"

Line 56. Add "respectively" after FtsZ

Line 71. Replace "mutations" with "and"

Line 79. Replace "some" with "most"

Line 91. Add "with" between "walls cells"

Line 106. Change "is switched" to "switches"

Line 105. Talking about cell division in L-forms. I think there should be some mention about the cell division in L-forms. Errington describes it as "membrane blebbing and tubulation". Also, growth of these cells is extremely slow (as seen in Ext Data Fig. 1). In other words, there should be some mention that L-form specific division is unusual at best and possible due to unregulated physical forces.

Line 135. I have trouble with the title of this section since L forms were made in E. coli by Errington's group in 2014. doi: 10.7554/eLife.04629

Line 136. The second sentence starting "In B. subtilis..." is not complete.

Line 143. I think Cef also inhibits PBP1a

Line 144-5. Move "less" to before "peptidoglycan"

Line 155. Add "respectively" at the end of the sentence

Line 173. Sentence starting "We measured.." Not sure what is meant here. Over what length of time were cells monitored and how long after the antibiotics were added.

Line 297. Add "we" before "expected".

Line 303. Shape should be shapes

Line 313. The sentence containing "...cell wall synthesis catalyzed only by PBP2 or PBP3..." is not correct. Cell wall synthesis catalyzed by PBP1s is also occurring.

Line 318-319. Change sentence to "Inhibition of the activity of PBP2 and PBP3 by Mec and Azt, respectively, allows cells to grow as L-forms."

Line 331-4 and 351-353. Although there has been a suggestion that the Rod complex was involved in division, the division of rodA or mreBCD cells argues that division occurs without the Rod complex. (work from many labs including de Boer). It should be mentioned that division can occur without the Rod system, at least with these deletion strains.

Lines 381-2. Exactly like rodA or mreBCD cells.

Line 389-91. This is known from rodA or mreBCD cells and this should be mentioned.

Reviewer #3

(Remarks to the Author)

The authors have addressed most of my concerns. However, I have a few additional minor points:

1. The sentence in lines 136-137 is incomplete and needs revision.
2. In line 233, the authors mention that they analyzed the localization of ZapA-GFP in three dimensions, as shown in Extended Data Fig. 3b. However, the z-stack images in that figure appear almost identical. Rather than displaying individual images from each z-stack, the authors should provide a reconstructed 3D ring structure.
3. The format of the references is inconsistent. Some references use full journal names, while others use abbreviations. Please ensure uniform formatting throughout.

Reviewers' comments:

Reviewer #1 (Remarks to the Author):

In this manuscript Shiomi and colleagues investigate the recovery of cells induced to
become L-forms. They develop a process so they monitor the dynamics of cell shape as
PG synthesis is inhibited and then allowed to resume (elongasome or divisome or both).
Basically, they show that distorted L-form cells can recover more uniform shape and size
by utilizing Z ring driven septal PG synthesis. Also, they show that Min and SImA contribute
to recovery by locating the Z ring near midcell. It is a nice demonstration that Z rings can
form in amorphous blobs, and if PG synthesis can occur, result in cell division, eventually
leading to more consistent size and shape.

Errington's lab has looked at recovery of L forms with and without FtsZ when PG synthesis
is allowed to resume. With FtsZ cell shape is recovered (both elongasome and divisome
are active) whereas without FtsZ (only elongasome active) cells can recover a rod shape
but die due to the absence of division.

This is somewhat like growing cells deleted for the elongasome where only septal PG
synthesis is occurring. In that case the Min system is known to be important (probably
SImA as well but I don't think it was known at the time and may not have been tested).
Also, knocking out the Rod system generally is lethal although suppressors occur at a high
frequency. It appears here that the media for growing L-forms probably suppresses such
lethality so suppressors are not needed.

One cautionary comment though is that the authors seem to overstate their case due to
probably unintended language. For example, on line 35 it is mentioned that control is
exerted in the absence of a cell wall. This is not strictly the case though as septal PG
synthesis is required. Thus, what is shown is that starting out with amorphous blobs
recovery of a more uniform shape can occur utilizing the cell division machinery (i.e. septal
PG synthesis driven by the Z ring). The title correctly summarizes the findings. I don't think
the findings are unexpected, like I said it is like growing cells without the elongasome, but
monitoring dynamics of the recovery reinforces what we know about cell division.

Also, Z rings do not appear (except for 9.5%) to be required for division in the absence of a
cell wall as shown in ref 39. But what do the authors think is going on in the 9.5% of cells
where FtsZ appears at constrictions. Is this mere coincidence?

***When antibiotics are removed, cells are constricted at the location where the Z-ring***
***is present and start to divide. On the other hand, in L-form cells, even though there***
***is a Z-ring in the division position (9.5% of cells in our experiment), there is no***
***constriction after cell wall synthesis or as seen in rod-shaped walled E. coli cells.***
***Therefore, we think that in the 9.5% of cells in which the localization of the Z-ring***
***coincides with the division position, nucleoid occlusion prevents the formation of Z***
***rings on top of nucleoids, resulting in the localization of the Z-ring between the***
***nucleoids, where division is observed incidentally.***

During the recovery from inhibition of cell wall synthesis is it possible to estimate the
efficiency of this process? It appears to be fairly efficient but hard to judge. What I am
asking is how much cell mass is lost and unable to recover.

***We measured how many L-forms observed at 12 hours could revert to rod-shaped***
***cells. As a result, differences were observed depending on which antibiotic was***
***used for the conversion to the L-form. These results are described in the text.***

***Fos : 4/132, 3.0%***

***PenG : 38/207, 18.4%***

***Cef : 37/172, 21.5%***

***Not all cells could revert to walled cells, especially Fos L-forms, which had a low***
***reversion efficiency.***

Line 57. Why not mention FtsW here since you bring in up a few lines later.

***We added a description of FtsW in the Introduction.***

Line 64. "Work in other bacteria, including... This wording could be changed since wall-less
bacteria lack most components.

***We revised the sentence as follows. "Therefore, it remains unknown whether the Z***
***ring also works in wall-less bacteria, such as Mycoplasma species, which lack most***
***divisome proteins other than FtsZ".***

Line 83. I think the role of FtsZ is controversial since deleting FtsZ from the 19 genes

added back did not affect cell size/shape much. Summarized in Trends in cell biology
Volume 32, Issue 11, November 2022, Pages 900-907

***We thank the reviewer for pointing this out. We have changed the notation from***
***syn3B to syn3+126, the smallest set of seven genes that speed up the growth rate of***
***syn3 and uniformly change its morphology. The ftsZ gene is included among these***
***seven genes (Pelletier et al., 2021).***

Line 135. I don't think E. coli growing as an L-form without FtsZ is controversial. Ref 39 is
very convincing that FtsZ is not required.

***Mercier et al., showed that E. coli L-forms converted by Fos did not require FtsZ for***
***growth, but Joseleau-Petit et al. showed that E. coli L-forms converted by Cef***
***required FtsZ for growth. Thus, we think that requirement of FtsZ for L-form growth***
***is controversial, and we would like to clarify this at first in this study.***

Sentence starting on line 141 is very long and could be split into two sentences.

***As the reviewer suggested, we split the sentence.***

Line 155-7. Ref 39 showed this was possible – i.e. for L-forms to grow without FtsZ

***We agree the comment and added the ref in the text.***

Line 176. Sentence starting with – These results... I cannot follow what you mean here.

The differences between the antibiotics is not clearly explained. Fos blocks at precursor

stage, Pen only blocks transpeptidation so glycan chains can form (may or may not be

stable), and with Cep, only the divisome is blocked.

***We added the explanation of the modes of action of antibiotics in the first paragraph***
***of “Results and Discussion” section to clarify the difference. We also changed the***
***sentence to “These results suggest that FtsZ is required to revert to walled cells***
***from cells completely lacking cell wall.”.***

Line 197. The 9.5% are interesting. Is FtsZ involved with these?

***As we answered in the first comment, we believe that the localization of FtsZ to the***
***division site is independent of the division of L-forms.***

Line 302-3. Note point in Line 83 comment above.

***As responded to in a previous comment, we have changed the notation from syn3B***
***to syn3+126, the smallest set of seven genes that speed up the growth rate of syn3***
***and uniformly change its morphology.***

Line 336. Corraling is the wrong word here. MinC prevents Z rings at poles but not at
midcell.

***As suggested, we changed the sentence “This inhibitory activity of MinC near the***
***cell poles helps to restrict Z rings at midcell by corralling aberrantly localized FtsZ***
***polymers,” to “This inhibitory activity of MinC near the cell poles helps to restrict Z***
***rings at midcell by preventing aberrantly localized FtsZ polymers,”.***

Line 382. This is consistent with Δ MreB cells grown with extra FtsZ or in minimal medium.
***As MreB is not discussed in this paper and the revised manuscript is more focused***
***on the min system and nucleoid occlusion, we decided not to include mreB here to***
***make the logic clear to the reader.***

Ext data fig. 10. How sure are the authors about the conservation in this figure. For
example, SlmA is in the TetR family. Are the authors sure they are picking up true
homologs and not paralogs? The names of the bugs are a bit faint in the figure.

***Thank you for your insightful comment. We understand your concern regarding the***
***distinction between orthologs and paralogs. We believe that our analytical approach***
***minimizes the impact of potential paralog over-estimation for the following reasons:***
***1. Stringent threshold: We employed a strict criterion of e-value < 1e-8 in our HMM***
***search, which significantly reduces false positives.***

***2. Hit count distribution: For most genes, the number of hits per genome is***
***concentrated at 0 (no hits) or 1 (single hit).***

***For example:***

***slmA: Out of 4,326 genomes, 3,365 had no hits, 874 had a single hit***

***MinC_C: 2,421 had no hits, 1,872 had a single hit***

***MinC_N: 3,186 had no hits, 1,140 had a single hit***

***3. Rarity of multiple hits: Genomes with two or more hits are very few (e.g., 87***
***genomes for *slmA*, 33 for *MinC_C*).***

***These results strongly suggest that paralog over-estimation is unlikely to***
***significantly affect the overall distribution or our conclusions.***

***We have provided a high-resolution version of the figure to improve the clarity of the***
***phylogenetic tree labels.***

Reviewer #2 & #3 (Remarks to the Author):

Examining the interplay between the cell size-determining mechanism and cell wall
synthesis poses a challenge due to the fundamental role of cell wall synthesis in walled
bacteria. In this study, Hayashi et al. utilized wall-less E. coli L-form cells and demonstrated
that FtsZ-dependent septal cell wall synthesis alone is sufficient to transform
heterogeneous cell morphology into predominantly uniform oval shapes. Additionally, they
elucidated that at least one of the Min or nucleoid occlusion systems is requisite for the
transition from FtsZ-independent division in wall-less cells to FtsZ-dependent division in
septal-walled cells. The manuscript is lucidly articulated and logically structured. I would
endorse its publication pending the resolution of the mentioned concerns.

Major Concerns:

1. The relationship between FtsZ ring constriction, septal cell wall synthesis, and cell
division in L-form cells is not entirely clear. Previous work suggested that cell wall synthesis
is the driving force for cell division in both E. coli (PMID: 26831086) and S. aureus (PMID:
36947615). Recent studies (PMID: 33495624, bioRxiv 515301) demonstrate that active
septal cell wall synthesis can occur independently of fast FtsZ treadmilling dynamics.
Therefore, it's crucial to determine whether FtsZ ring constriction triggers septal cell wall
synthesis or if septal cell wall synthesis facilitates FtsZ ring constriction. Consequently, the
statement regarding "FtsZ-dependent septal cell wall synthesis" should be carefully
rephrased.

***We agree with this point. We replaced “FtsZ-dependent septal cell wall synthesis”***
***with “FtsZ-dependent division” in the Summary section.***

2. The authors state that FtsZ forms ring structures in E. coli L-form cells, which is distinct
from B. subtilis L-forms. Given that a ring is a 3D structure not easily represented in 2D
projections, could the authors employ 3D imaging techniques such as optical sectioning
confocal microscopy, 3D-SIM, or 3D-SMLM to visualize the complete Z-ring structure?

**We added the data showing 3D of the Z ring in L-forms in Extended Data Fig. 3.**

3. The primary findings of this study are derived from observations of cell morphology. To
bolster their conclusions, the authors should incorporate growth curves to provide evidence
of cell division or lysis.

**E. coli L-forms hardly grow in liquid medium in test tubes unlike B. subtilis L-forms.**
**Thus, we cannot include the growth curve of the E. coli L-form cells.**

Minor Concerns:

1. It is advisable for the authors to conduct Western blotting to assess the residual amount
of FtsZ under FtsZ-depletion conditions.

**We added the data showing the amount of FtsZ protein in the depletion strain in**
**Extended Data Fig. 1.**

1. In Fig. S1, it would be beneficial to include growth curves of L-form cells after removing
antibiotics instead of just showing some representative images.

**As we responded above, we are unable to include the growth curve in this figure**
**because E. coli L-forms do not grow in liquid in test tubes.**

2. The authors should compare the percentage of L-form cells with Z-rings coinciding with
the division position under different antibiotic treatment conditions. If, as the authors
suggest, "the constriction of the Z-ring requires cell wall synthesis," there should be a
higher coincidence of Z-rings with the division position in cells treated with Cef compared to
those treated with PenG or Fos.

**We measured cells localizing the Z ring at the division site in L-forms. In 9.5%**
**(16/169 cells) in Fos-treated L-forms and 9.5% (16/169 cells) in PenG-treated L-forms,**
**cells localized the Z-ring to the division site. In contrast, a slightly higher percentage**

**(12.5%, 7/56 cells) was observed in Cef-treated L-forms, possibly due to the fact that**
**a little cell wall remained in Cef-treated L-forms. However, in any case, no clear**
**constriction was observed at these division sites, which could be due to the Z-ring**
**tending to localize in the space between nucleoids. This was described in the text.**

3. Might the authors offer additional insights into the factors underlying the assembly of Z-
rings in E. coli L-forms compared to their absence in B. subtilis L-forms, taking into account
factors beyond the distinct MinC-MinD dynamics?

**One possibility to account for the difference between E. coli and B. subtilis would be**
**the different activities of SlmA in E. coli and Noc in B. subtilis on the Z ring. We**
**described in the text as follows. "In addition, in E. coli SlmA interacts directly with**
**FtsZ filaments to cause depolymerization ⁸⁴, whereas in B. subtilis Noc, an important**
**protein for nucleoid occlusion in B. subtilis, binds to DNA and the cell membrane,**
**physically inhibiting FtsZ filament formation in the vicinity of the membrane ⁸⁵. This**
**difference in MinC-MinD dynamics and the different functions between SlmA and**
**Noc may account for differences in the ability of E. coli and B. subtilis to form Z**
**rings in L-form cells."**

4. In Fig. 3a, it is noteworthy that a majority of cells lysed after 12 hours when treated with
Mec and Cep. Could the authors measure the growth curve to demonstrate that the cells
are still actively growing and dividing?

**As we responded above, it is difficult to measure the growth rate of E. coli L-forms,**
**so we cannot include the data. Cell death was observed in L-form cells incubated for**
**12 hours in the presence of the antibiotic Fos, Mec and Azt (Fig3a, b, Extended Data**
**8), comparable to that observed with only Fos treated (Extended Data 2a). On the**
**other hand, we considered that these cells continue to grow and divide because**
**cells continue to divide as L-forms even under conditions in which Mec or Azt is**
**subsequently removed (Extended Data 8a, b, Movie11).**

5. In Fig. 3b, the authors observe that E. coli SWD cells gradually convert into uniform oval
shapes after the removal of Cep, and state "This indicates that cell shape can be controlled
only by septal cell wall synthesis and not by the cylindrical cell wall." This conclusion may

be too definitive, as the cylindrical cell wall likely plays a significant role in determining cell
shape.

***We agree the point and changed the sentence to “This indicates that cell shape can
be controlled solely by septal cell wall synthesis even if the cylindrical cell wall is
not synthesized”.***

6. In Fig. 3c, could the authors also consider imaging ZapA-GFP to determine if the FtsZ-
ring co-localizes with HADA signal?

***We tried several times to stain cells with HADA but the results were not clear. So, we
used GFP-FtsN^{SPOR} to show septal cell wall because it localizes specifically to
denuded septal peptidoglycan. As expected, we observed that GFP-FtsN^{SPOR}
localized diffusely in the periplasm in L-forms while it localized to the septum after
the removal of Fos and Azt (Fig. 3C and Extended Data Fig. 9).***

7. Fig. 2 demonstrates that FtsZ (indicated by ZapA-mCherry) co-localizes with FtsN in L-
form cells. However, in Fig. 4 and Fig. S7, both ZapA-GFP and GFP-FtsN display a mesh-
like structure in Δ minC L-form cells. Notably, ZapA-GFP appears more punctate compared
to the smoother mesh observed with GFP-FtsN. Does this imply that FtsZ is not co-
localized with FtsN in Δ minC L-form cells?

***We thank the reviewer for this comment. We added descriptions about the role of
MinC for formation of the filaments in L-forms. FtsN should not form rings / filaments
independent of Z rings, so that we think the smooth filaments pointed out by the
reviewer may be due to the slight overproduction of GFP-FtsN expressed from a
plasmid.***

8. Fig. 6 is a pivotal figure summarizing the model. It is mentioned only once incidentally in
the main text. Could the authors add a sentence providing further description of this model
in the main text?

***As the reviewer suggested, we added sentences to explain Fig. 6 in the section “Cell
shape of SWD cells lacking minC or slmA”.***

9. Fig. S10 appears too small to discern details clearly. Consider enhancing the figure for
better visibility.

***We have provided a high-resolution version of the figure to improve the clarity of the***
***phylogenetic tree labels.***

10. The format of references should be consistent.

***We revised the references.***

Reviewer #4 (Remarks to the Author):

The authors studied cell shape and division phenotypes of E .coli after treatment with
various drugs that interfere with all, or specific sets of, peptidoglycan (cell wall) synthases,
in a nice experimental set-up that allows for long-time monitoring of growing walled or wall-
less cells in microfluidic devices

However, though treatment of cells with Fos, PenG, and maybe Cef are expected to lead to
cells that are mostly wall-less or L-form, there is no compelling reason to assume (as the
authors do) that cells treated with both Mec and Cep will be without PG. Two major PG
synthases (the preferred targets of Cef), PBP1A and PBP1B, are expected to still produce
PG under those conditions. They may look amorphous and wall-less, but this is likely
caused, at least in part, by the confined space in the microfluidic device. Thus, without
convincing evidence showing otherwise, these cells should not be referred to as L-forms.

As both the title, abstract and main conclusions of the manuscript are mainly based on the
work with such (Mec + Cep treated) cells, from which then one of the drugs is often
withdrawn (leading to the cells named SWE or SWD by the authors), the interpretation of
the results by the authors is not convincing and likely wrong. Several other assertions by
the authors are debatable as well.

***We understand the concerns of this reviewer and we agree. Therefore, we added Fos***
***as well as Mec and Azt (we changed Cep to Azt in the revised experiments as the***
***reviewer suggested) to convert cells to L-forms that should completely lack cell wall,***
***because Fos inhibits MurA activity that is required for the first step of peptidoglycan***
***synthesis.***

Specific remarks

1) Lines 146-154. The authors find that wt and FtsZ-depleted cells can propagate as L-form
on NA/MSM agar plates containing Fos, PenG, or Cef (Fig. S1a, as expected), and content
that they do so in the microfluidic device as well (line 153).

a) Perhaps PenG or Cef-treated cells do so, but the pictures (Fig. S1b) and movies (1, 2, 7,
8) shown for Fos-treated cells are not convincing. Most cells seem to die/lyse. Remaining
cell structures can change shape, but few, if any, actual 'division' events are evident before
removal of the drugs. How do you know these cells actually propagate rather than a few
cells barely surviving?

***We revised Extended Data Movie7 to follow the conversion of a single cell from***
***walled to L-form and back to walled again. We measured how many WT or Δ ftsZ L-***
***forms divided in the presence of Fos, PenG, or Cef. All of the L-form cells could***
***divide independently of FtsZ. We added the results in the text.***

***WT (Fos): 21.2% (22/104)***

***WT (PenG): 55.7% (39/70)***

***WT (Cef): 30.4% (38/125)***

***Δ ftsZ (Fos): 22.0% (26/118)***

***Δ ftsZ (PenG): 33.3% (36/108)***

***Δ ftsZ (Cef): 44.2% (53/120)***

b) It is also interesting to note that most 'division' events in PenG or Cef-treated cells seem
to correlate with cells that appear to be moving in a medium stream either hitting the
support braces of the device, or being extended and 'pulled apart' within this stream (see
movies, before drug withdrawal). This suggests that 'division' of these cells is driven by
contact and/or sheer forces within the device. In how far this is comparable to agar-grown
L-form cells is unclear.

***As the reviewer pointed out, cells appeared to divide by hitting the support braces of***
***the device or by the flow of the medium. Therefore, we performed a similar***
***experiment with the pressure of the medium set to zero on the program driving the***
***device. However, the flow did not disappear under those conditions (0.1-0.3 μ L/h).***
***Normally, the experiment is carried out at a flow rate of 2.5 μ L/h, so setting the***
***pressure to zero brought the flow as close to zero as possible. Z ring-independent***
***division was observed in some cells when the experiment was performed at the***
***slowest flow velocity achievable with this apparatus. We have described the results***

***in the text and made a new figure (Extended Data Fig. 3e) and movie (Extended Data***
***Movie9).***

2) Line 171-172. The authors state that PenG-treated L-forms produce less PG than Cef-
treated ones. As pointed out in ref 40 (but not here), however, the evidence for this is quite
debatable as HADA incorporation itself may be less efficient in the former. This might also
be the case for Fos-treated cells, by the way. The only way to know for sure is to actually
extract and measure the amount of PG in the treated cells. This seems especially relevant
to the Mec + Cep-treated cells (see below).

***As responded to the reviewer's comment below, we used Fos as well as Mec and Azt***
***(which replaced Cep) (please see our above response).***

3) Lines 176-177. The statement 'These results suggest that FtsZ is required to trigger the
first step of peptidoglycan synthesis.' is incompatible with the finding that delta-ftsZ cells
can grow as walled coli-flowers (ref 39).

***Mercier et al., showed that Δ ftsZ cells failed to form L-form colonies. They also***
***showed that if Δ ftsZ cells acquired suppressor mutations such as mrcB, lpp, and***
***wcaJ mutations, the cells formed coli-flower colonies. Thus, without any suppressor***
***mutations, Δ ftsZ cells can't form colonies. Therefore, we believe our results are***
***consistent with the previous results (Mercier et al., 2016).***

4) Lines 177-180. I don't understand this statement. In what way(s) are your results
consistent with those findings in ref 41? Please explain to the reader.

***We revised the sentence to*** "The results resemble the observation that cell division
precedes cell elongation during reversion from spheroplasts without peptidoglycan."

5) Lines 182-192.

a) That E.coli L-forms can propagate without ftsZ was first shown by Mercier et al (Elife,
2014), and this reference should be added here.

***We added the correct reference (Mercier et al., 2016, Nature Micro) here.***

b) Lines 183-185. The logic of this statement escapes me. Please explain to the reader.

**To clarify this, we revised the sentence to** “we predicted that, in *E. coli* L-forms, FtsZ
might form non-ring filamentous structures similar to those observed in *B. subtilis* L-
forms³²”.

c) Line 85 implies that *B. subtilis* L-forms do make Z-rings, while lines 189-190 imply the
opposite. Which is it?

**Z rings are not formed in *B. subtilis* L-forms but are present in *E. coli* L-forms.**

401 d) Also, please add the appropriate reference(s) concerning FtsZ localization in *B. subtilis* L-
402 forms.

**We added the ref to the sentence** “we predicted that, in *E. coli* L-forms, FtsZ might
form non-ring filamentous structures similar to those observed in *B. subtilis* L-forms³²”.

6) Lines 198-200. Perhaps this statement can be further solidified; did you see any
constriction, without actual division, at any of the rings at all?

**As already responded to the other reviewer’s comments, we checked the localization
site of Z rings and division sites.**

**Walled cells are constricted at the location where the Z-ring is present and start to
divide. On the other hand, in L-form cells, even though there is a Z-ring in the
division position (9.5% of cells in our experiment), there is no visible constriction
after cell wall synthesis or as seen in rod-shaped walled *E. coli* cells. Therefore, we
think that in the 9.5% of cells in which the localization of the Z-ring coincides with
the division site, nucleoid occlusion prevents the formation of Z rings on the
nucleoid, resulting in the localization of the Z-ring between the nucleoids, where
division is observed incidentally.**

7) Lines 207-211. This statement is not supported by the data. The cell shown in Fig.1a is
from 9 hr after Fos withdrawal. How likely is it that this ring persisted for 9 hr?

**We revised Extended Data Fig. 4 to show that the Z ring persisted during the
observation.**

8) Lines 223-224.

a) A few 'what' were localized to the cell pole?

b) Also, I don't see any polar GFP-PBP3 in fig. S3a. There is a focus at a quarter position.

**According to the suggestion, we revised the sentence to "a few dot-like foci of GFP-**
**PBP3 were localized to the cell peripheries".**

9) Lines 226-229. Fig. 2b.

The localization of GFP-FtsN looks pretty sharp and it would be interesting to know if this

depended on its N-terminal and/or C-terminal domain, which are weak and strong

localization determinants, respectively, in walled cells. As the C-terminal SPOR domain

binds septal PG, it is expected to not be required for this localization, but only if these cells

truly made no PG (see also point 2 above).

**We are grateful to the reviewer for suggesting experiments that will help to clarify**

**our conclusion. We constructed GFP-FtsN^{SPOR} and GFP-FtsN^{ΔSPOR} expressed**

**plasmids. In L-form cells, GFP-FtsN^{ΔSPOR} was localized to the Z ring but GFP-**

**FtsN^{SPOR} was not. Therefore, we concluded that cell walls were not synthesized on Z**

**ring in L-form cells. We added the results as a new Extended Data Fig. 6.**

10) Lines 241-244. Cells treated with Mec and Cep

a) How do you know that these cells divided independently of the Z-ring? Did you do the

same experiment with FtsZ-depleted cells? Please explain the evidence for this statement

to the reader.

b) Are these true L-form cells? PBP1A and 1B are still expected to be active and produce

PG. This is actually an important point. Wouldn't these cells just form large walled spheres

in liquid, if they were not trapped by the low ceiling in the microfluidic device? And, if not,

why not?

**In the revised version, we used Fos in addition to Mec and Azt (we used Azt instead**

**of Cep to inhibit PBP3 as the reviewer suggested) and got the same results as when**

**we used Mec and Cep (in the previous version of the manuscript). By using Fos, cell**

**wall synthesis is completely blocked and we can eliminate the effects of PBP1A and**

**PBP1B activities. Therefore, we thought that using Fos was better for our**

**experiments. We showed that Fos-L-forms divided independently of the Z ring in this**
**study.**

c) A related question is : What are the phenotypes of cells if they are treated with only Mec
(or Cep) in this device and medium?

**We added the data about the phenotypes of cells treated with only Mec as new**
**Extended Data Fig. 7 and Extended Data Movie10.**

470 d) Why did you choose such high concentrations of Mec and Cep (300 microgram/ml, line
420)? At this concentration neither antibiotic is likely very specific for their intended targets
anymore. Did you try lower concentrations?

e) Also, why did you choose cephalexin? Aztreonam is more specific for PBP3, and 2-3
microgram/ml is typically sufficient to inactivate it.

**We initially considered that a low concentration of antibiotics made**
**suppressors/revertants more likely to arise. As the reviewer suggested, we used**
**lower concentrations of Mec (10 µg/ml) and revertants were not observed. Therefore,**
**we replaced the previous results with the new one using 10µg /ml Mec. And in**
**response to comment 10e, we replaced Cep with Azt (20 µg/mL).**

11) Lines 249-251. The SWE cells are said to elongate (and swell). I actually see very little
elongation in the sense of rod-shape elongation in Fig. 3a or movie 10. Why don't these
cells revert to typical filament formation? Does the high Cep concentration perhaps also
affect PBP2 activity?

**After conversion to L-form cells with Fos, Azt and Mec, Fos and Mec were removed**
**from the medium, and SWE cells were not converted to filamentous cells but some**
**cells showed coli-flower-like morphology (Fig. 3a and Extended Data Movie12). The**
**reversion to rod-shaped cells, even though they are filamentous, may require FtsZ**
**(see lines 421-422 of the revised manuscript). As we used Azt instead of Cep, it is**
**less likely that PBP2 activity is inhibited after Foc and Mec are removed, although it**
**cannot be completely ruled out.**

12 Lines 253-258. SWD cells.

a) The conclusion that PBP2 activity is not needed for cell constriction is certainly not new.
It has long been known that none of the rod-system proteins, including PBP2, are strictly
required for cell constriction.
**According to the suggestion, we modified the sentence to “This suggested that**
**activation of PBP3 is sufficient to initiate Z ring constriction and cell division in SWD**
**cells. Although it was previously thought that cell wall synthesis for division is**
**triggered by the interaction between the Rod complex containing PBP2 and the**
**divisome containing PBP3, PBP2-mediated synthesis of cylindrical cell wall is not**
**required for constriction and division.”.**
b) However, at high growth rates and/or rich medium, rod-system mutants typically form
(very) large non-dividing walled spheres unless the cellular level of FtsZ is modestly
elevated, in which case they can propagate as smaller dividing walled spheres. One
surprising aspect of Fig. 3b and Movie 11, is that the SWD spheres (or 'ovals') at the end of
the experiment are so very tiny, especially compared to the size of the rod-shaped cells at
the beginning of the experiment. Why are they so tiny?
**Although we do not have the answer for the comment at this moment, in our**
**microfluidic device, the reverted cell size was limited by the ceiling height of 0.7 μm .**
**This may cause cells to be small.**
c) What is the estimated doubling time of wt cells without drugs in this device and medium?
**We calculated the doubling time of WT cells without drugs. The doubling time was**
**about 40 min in NBMSM in our microfluidic device, which was slightly slower than**
**that in liquid culture.**
524 d) If you grow an mrdA (no PBP2) mutant under these anaerobic conditions, are they also
this tiny?
**We think that small cells are caused by the height of the ceiling (0.7 μm) as we**
**mentioned above. Cells with inhibited PBP2 activity are larger than WT cells (Shiomi**
**et al., 2008). Thus, we expect that ΔmrdA or cells with no PBP2 activity would**
**become smaller in our device.**

e) Do these growth conditions perhaps elevate the cellular FtsZ concentration?

***There is a possibility that FtsZ concentration is increased in SWD cells. However, as***
***we cannot take the cells out of the device, we cannot answer this question.***

13) Line 259. The HADA experiment does not confirm that 'cell wall synthesis was re-
initiated only at the Z ring constriction in SWD cells'. Brighter HADA staining at division
sites is also seen in wt cells and does not mean PG synthesis is exclusive to these sites. In
fact, one expects PBP1A and 1B to produce PG all along the periphery of the SWD cells.
See point 10 as well.

***HADA staining experiments were difficult to stably reproduce results. Therefore, we***
***examined localization of GFP-FtsN^{SPOR} in SWD cells. GFP-FtsN^{SPOR} was localized***
***uniformly in L-form cells, presumably in the periplasm, but was localized to the***
***division site in SWD cells (Fig. 3, Extended Data Fig. 6). This localization coincided***
***with ZapA-mCherry (Extended Data Fig. 9). Therefore, cell division in SWD cells was***
***dependent on the Z ring and septal wall synthesis. We added these results as new***
***Fig. 3 and Extended Data Fig. 9.***

14) Lines 271-272. The authors conclude: ' However, our results show that FtsZ-dependent
cell division occurs in L-form cells and does not require the synthesis of the cylindrical cell
wall.'

I disagree, because I don't believe these are L-form cells, but rather comparable/equivalent
to walled rod-system mutant cells (such as delta-mrdA). See points 10 and 12, above.

***Please see our response to the 10 (a) and (b).***

15) Lines 284-306, Fig. S5, Movies 11 and 12.

a) This whole section reinforces my argument (points 10, 12-14, above). These SWD cells
behave like a rodA (mrdB) mutant because they basically are equivalent, except that these
are essentially wt cells with PBP2 inactivated (so, more like a mrdA mutant, but the
phenotype is similar).

***We treated cells with Fos, Mec, and Azt to convert them to L-forms. The cells were***
***still amoeboid even when only Fos was removed from this medium. PBP1A and***
***PBP1B may be active in this state as the reviewer stated, but the cells remained***

***amoeboid and did not transform into an oval shape. Further exclusion of Mec from***
***here (i.e., only Azt was present) did not change the cell morphology. On the other***
***hand, when Fos and Azt were removed from the medium containing Fos, Mec, and***
***Azt, and only Mec was present, the cells deformed into an oval shape. Thus, when***
***only the activity of PBP3 or PBP2 is inhibited from L-forms that are not completely***
***covered by the cell wall, the cells become oval-shaped when only PBP2 is inhibited,***
***but they remain amoeboid-like when only PBP3 is inhibited. If the entire cell is***
***covered by the cell wall, then inhibiting only PBP3 should elongate the cell, but this***
***did not happen. The state of the cell must be essentially different from when the cell***
***wall is synthesized by activating only PBP2 from a state in which there is no cell wall***
***at all. In other words, we can conclude that our SWD cells are different from mrDA***
***cells.***

b) Just take wt cells, put them in the device and add medium with Mec. You'll end up with
the same 'oval' cells. See point 10c as well.

***As we responded in the above comment, we added the data about cells that were***
***treated with only Mec (Extended Data Fig. 7 and Extended Data Movie10). These***
***cells became round in the microfluidic device.***

16) Fig. S6. I assume growth conditions here were aerobic? Please state in the legend.

***According to the suggestion, we modified the all of figure legends.***

17) Lines 308-351. The results in this section are nice but, as indicated by the authors, not
very surprising given what we know about FtsZ and the Min and SlmA systems, and
previous results with mre mutants and lipid vesicles.

***We think this comment is related to your comment (19-a). Please see our response to***
***comment (19-a).***

18) Lines 363-364. I don't believe SWD cells produce PG only at midcell as stated here and
elsewhere. See points made above.

***Please see the above response (10a and 10b).***

19) Lines 374-384. delta-minC delta-slmA SWD cells, Fig.5a, Fig. S8, Movie 13.

a) I predict that you'll see the same phenotype if you just grow the RU2409(minC slmA)
cells in NB/MSM medium in batch culture (as in Fig. S6) or in the microfluidic device, but
now with the addition of Mec. The many filamentous FtsZ structures that are allowed to
form at random positions in the resulting walled spheres (batch) or squashed walled
spheres (microfluidic device) in the absence of both MinC and SlmA compete with each
other for division proteins such that assembly of a complete/functional ring is suppressed.
Quite similar to what happens when delta-minC delta-slmA cells are grown in rich medium,
and cells filament.

***Bailey et al. 2014 predict that there will be a factor(s) other than the Min system,***
***nucleoid occlusion, and MatP-mediated Ter linkage that determine the localization of***
***the Z ring to the dividing site. In their experiments, the Z ring could still localize to***
***the cell center in Δ min Δ slmA Δ matP cells. Perhaps the cell wall synthesis of the***
***cylindrical portion plays an important role in the localization of the Z-ring, whether***
***directly or indirectly is unknown. On the other hand, even round-shaped walled-cells***
***such as Δ mrda cells can divide in the center of the cell. Based on the above, it is***
***likely that the min system, nucleoid occlusion, Ter linkage, and cell wall synthesis all***
***contribute to some extent, and that the localization of the Z-ring is more than***
***complete when all of them are gone. The following text has been added to the text to***
***discuss this point.*** “Previously, it was shown that the Z ring localizes to the division
site independently of the Min system, nucleoid occlusion system, and the Ter-linkage in
walled *E. coli* cells⁸⁶. Considering their data together with our data, it is possible that
cell wall synthesis in the cylindrical portion may facilitate the localization of or stabilize
the Z ring to the division site.”

b) Oddly, Fig.S9 and Movie 14 are not mentioned/discussed. What is the point of this data?

***We added the explanations for the figure and movie.***

20) Fig.6 is also ignored in the text.

a) What is the point of the figure?

***We added the description of fig6 in the revised version (line 467-489).***

b) One of the (f) panels is mislabeled.

***We corrected and revised the figure.***

21) Lines 183-500. Please indicate what ceiling height in the microfluidic device was used
(0.7 micron throughout?)

***All the experiments were done with a 0.7 μm ceiling height. We modified the methods***
***section.***

22) Genetic descriptions of strains and/or strain names in the legends of figures and
movies are often imprecise. Please check and correct.

***We checked and corrected the mistakes throughout the manuscript.***

Reviewer #1 (Remarks to the Author):

This revision is improved. The main conclusion is supported by additional experiments. The conclusion is that Z rings can form in amoeba-shaped E. coli L-forms and a complete divisome is assembled (FtsN is localized but SPOR is not-indicates a complete divisome is formed but not active). However, the Z ring does not constrict in the absence of PG synthesis. If septal PG synthesis occurs (PBP3 active), then division occurs in an FtsZ-dependent manner and amoeba-like cells are eventually converted to a more uniform coccal-shape. It is already known that $\Delta rodA$ or $\Delta mreBCD$ cells grow as coccal-shaped cells under slow growth conditions and that Min has an important role. The additional information here is that Z rings (and complete divisomes) form in the amoeba-like L-forms but don't constrict. However, if septal PG synthesis is allowed to occur, they constrict.

These experiments don't answer some of the big questions that are asked by the authors such as the role of FtsZ in wall-less cells. It is still not clear what the role of FtsZ is in wall-less mycoplasma. Clearly FtsZ has evolved in E. coli to be coupled to septal PG synthesis.

Some suggestions for improvement:

Line 47. Pressures does not need to be plural.

We changed "pressures" to "pressure". (line 46)

Line 51. "control" should be "controls"

We changed "control" to "controls". (line 50)

Line 56. Add "respectively" after FtsZ

We added "respectively" after FtsZ. (line 55)

Line 71. Replace "mutations" with "and"

We reconfigured this sentence for clarity. (line 69-70)

Line 79. Replace “some” with “most”

We replaced “some” with “most”. (line 78)

Line 91. Add “with” between “walls cells”

We added “with” between “walls cells”. (line 89)

Line 106. Change “is switched” to “switches”

We changed “is switched” to “switches”. (line 110)

Line 105. Talking about cell division in L-forms. I think there should be some mention about the cell division in L-forms. Errington describes it as “membrane blebbing and tubulation”. Also, growth of these cells is extremely slow (as seen in Ext Data Fig. 1). In other words, there should be some mention that L-form specific division is unusual at best and possible due to unregulated physical forces.

We added the sentence “Whether FtsZ is required for the division of *E. coli* L-forms is controversial. D’Ari’s lab showed that FtsZ is required for L-form growth in the presence of cefsulodin³⁵ whereas Errington’s lab showed that cell division depends on membrane fluidity and is independent of the Z ring machinery (divisome) in the presence of fosfomycin^{36,37}. If L-form division is indeed independent of FtsZ, then division would likely be driven by unregulated physical forces such as membrane blebbing and tubulation.”. (line 104-109)

Line 135. I have trouble with the title of this section since L forms were made in *E. coli* by Errington’s group in 2014. doi: 10.7554/eLife.04629

We changed the title of the section to “Confirmation of the non-essentiality of FtsZ in *E. coli* L-form cells.” because the essentiality of FtsZ in *E. coli* L-forms is controversial as written in the Introduction and this section. (line 140)

Line 136. The second sentence starting “In *B. subtilis*...” is not complete.

We revised the sentence to “In *B. subtilis* L-form cells, FtsZ forms filaments instead

of discrete rings³².” (line 141-142)

Line 143. I think Cef also inhibits PBP1a

We revised the sentence: “Cef inhibits the transpeptidase activities of PBP1A and PBP1B⁴⁷.” (line 148)

Line 144-5. Move “less” to before “peptidoglycan”

We moved “less” to before “peptidoglycan”. (line 150)

Line 155. Add “respectively” at the end of the sentence

We added “respectively” at the end of the sentence. (line 161)

Line 173. Sentence starting “We measured..” Not sure what is meant here. Over what length of time were cells monitored and how long after the antibiotics were added.

To clarify the quantification, we revised the sentence to We first counted the number of cells 4 hours after the addition of antibiotic (Fos, PenG, or Cef). Of those cells, we counted the number of cells that were able to divide up to 12 hours later.” (line 179-181)

Line 297. Add “we” before “expected”.

We added “we” before “expected”. (line 304)

Line 303. Shape should be shapes

We changed “shape” to “shapes”. (line 310)

Line 313. The sentence containing...cell wall synthesis catalyzed only by PBP2 or PBP3... is not correct.

Cell wall synthesis catalyzed by PBP1s is also occurring.

We agree with the comment. We revised the sentence to “Next, after conversion to L-form cells with Fos, Mec and Azt, only Mec or Azt was removed from the media, re-

initiating cell wall synthesis for elongation or division in L-form cells.” (line 319-321)

Line 318-319. Change sentence to “Inhibition of the activity of PBP2 and PBP3 by Mec and Azt, respectively, allows cells to grow as L-forms.

We changed the sentence to “This suggests that inhibition of the activity of PBP2 and PBP3 by Mec and Azt, respectively, allows cells to grow as L-forms.” (line 326-328).

Line 331-4 and 351-353. Although there has been a suggestion that the Rod complex was involved in division, the division of $\Delta rodA$ or $\Delta mreBCD$ cells argues that division occurs without the Rod complex. (work from many labs including de Boer). It should be mentioned that division can occur without the Rod system, at least with these deletion strains.

We added two sentences to mention it.

“This resembles the division of *E. coli* cells lacking *rodA* or *mreBCD* in which the activity of the Rod complex is inactivated or decreased.” (line 343-344).

“This mode of division is similar to that of *E. coli* cells lacking *rodA* or *mreBCD* as described above.” (line 365-366)

Lines 381-2. Exactly like $\Delta rodA$ or $\Delta mreBCD$ cells.

We added “like $\Delta rodA$ and *S. aureus* cells” after “with each division”. (line 392)

Line 389-91. This is known from $\Delta rodA$ or $\Delta mreBCD$ cells and this should be mentioned.

We revised the sentence to “These results strongly indicate that cells can maintain a round or oval morphology even if they can divide in an FtsZ (or its functional equivalent protein) dependent manner without elongation like $\Delta rodA$ and $\Delta mreB$ cells.” (line 400-403)

Reviewer #3 (Remarks to the Author):

The authors have addressed most of my concerns. However, I have a few additional minor points:

1. The sentence in lines 136-137 is incomplete and needs revision.

We revised the sentence to “In *B. subtilis* L-form cells, FtsZ forms filaments instead of discrete rings³².” (line 141-142)

2. In line 233, the authors mention that they analyzed the localization of ZapA-GFP in three dimensions, as shown in Extended Data Fig. 3b. However, the z-stack images in that figure appear almost identical. Rather than displaying individual images from each z-stack, the authors should provide a reconstructed 3D ring structure.

To clearly show the three-dimensional structure of ZapA-GFP in L-forms, we showed one of the Z stacks and projective images of deconvolved images of ZapA-GFP in L-forms (Supplementary Fig. 3c). We also show the movie of deconvolved images as Supplementary Mov. 10.

3. The format of the references is inconsistent. Some references use full journal names, while others use abbreviations. Please ensure uniform formatting throughout.

We checked and revised the references.